# Deep learning is adaptive to intrinsic dimensionality of model smoothness in anisotropic Besov space

**Taiji Suzuki**
Department of Mathematical Informatics, The University of Tokyo, Tokyo, Japan
RIKEN Center for Advanced Intelligence Project, Tokyo, Japan
`taiji@mist.i.u-tokyo.ac.jp`

**Atsushi Nitanda**
Kyushu Institute of Technology, Fukuoka, Japan
RIKEN Center for Advanced Intelligence Project, Tokyo, Japan
`nitanda@ai.kyutech.ac.jp`

## Abstract

Deep learning has exhibited superior performance for various tasks, especially for high-dimensional datasets, such as images. To understand this property, we investigate the approximation and estimation ability of deep learning on *anisotropic Besov spaces*. The anisotropic Besov space is characterized by direction-dependent smoothness and includes several function classes that have been investigated thus far. We demonstrate that the approximation error and estimation error of deep learning only depend on the average value of the smoothness parameters in all directions. Consequently, the curse of dimensionality can be avoided if the smoothness of the target function is highly anisotropic. Unlike existing studies, our analysis does not require a low-dimensional structure of the input data. We also investigate the minimax optimality of deep learning and compare its performance with that of the kernel method (more generally, linear estimators). The results show that deep learning has better dependence on the input dimensionality if the target function possesses anisotropic smoothness, and it achieves an adaptive rate for functions with spatially inhomogeneous smoothness.

## 1 Introduction

Based on the recent literature pertaining to machine learning, deep learning has exhibited superior performance in several tasks such as image recognition (Krizhevsky et al., 2012), natural language processing (Devlin et al., 2018), and image synthesis (Radford et al., 2015). In particular, its superiority is remarkable for complicated and high-dimensional data like images. This is mainly due to its high flexibility and superior feature-extraction ability for effectively extracting the intrinsic structure of data. Its theoretical analysis also has been extensively developed considering several aspects such as expressive ability, optimization, and generalization error.

Amongst representation ability analysis of deep neural networks such as universal approximation ability (Cybenko, 1989; Hornik, 1991; Sonoda & Murata, 2017), approximation theory of deep neural networks on typical function classes such as Hölder, Sobolev, and Besov spaces have been extensively studied. In particular, analyses of deep neural networks with the ReLU activation (Nair & Hinton, 2010; Glorot et al., 2011) have been recently developed. Schmidt-Hieber (2020) showed that the deep learning with ReLU activations can achieve the minimax optimal estimation accuracy to estimate composite functions in Hölder spaces by using the approximation theory of Yarotsky (2017). Suzuki (2019) generalized this analysis to those on the *Besov space* and the *mixed smooth*

35th Conference on Neural Information Processing Systems (NeurIPS 2021).

Table 1: Relationship between existing research and our work. $\beta$ indicates the smoothness of the target function, $d$ is the dimensionality of input $x$, $D$ is the dimensionality of a low-dimensional structure on which the data are distributed, and $\widetilde{\beta}$ is the average smoothness of an anisotropic Besov space (Eq. (1)).

| Function class | Hölder | Besov | mixed smooth Besov | Hölder on a low-dimensional set | anisotropic Besov |
|---|---|---|---|---|---|
| Author | Schmidt-Hieber (2020) | Suzuki (2019) | Suzuki (2019) | Nakada & Imaizumi (2020); Schmidt-Hieber (2019); Chen et al. (2019) | This work |
| Estimation error | $\tilde{O}(n^{-\frac{2\beta}{2\beta+d}})$ | $\tilde{O}(n^{-\frac{2\beta}{2\beta+d}})$ | $\tilde{O}\Big(n^{-\frac{2\beta}{2\beta+1}} \times \log(n)^{\frac{2(d-1)(u+\beta)}{1+2\beta}}\Big)$ | $\tilde{O}(n^{-\frac{2\beta}{2\beta+D}})$ | $\tilde{O}(n^{-\frac{2\widetilde{\beta}}{2\widetilde{\beta}+1}})$ |

*Besov space* by utilizing the techniques developed in approximation theories (Temlyakov, 1993; DeVore, 1998). It was shown that deep learning can achieve an *adaptive approximation* error rate that is faster than that of (non-adaptive) linear approximation methods (DeVore & Popov, 1988; DeVore et al., 1993; Dũng, 2011), and it outperforms any linear estimators (including kernel ridge regression) in terms of the minimax optimal rate.

From these analyses, one can see that the approximation errors and estimation errors are strongly influenced by two factors, i.e., the *smoothness* of the target function and the *dimensionality* of the input (see Table 1). In particular, they suffer from the *curse of dimensionality*, which is unavoidable. However, these analyses are about the worst case errors and do not exploit specific intrinsic properties of the true distributions. For example, practically encountered data usually possess low intrinsic dimensionality, i.e., data are distributed on a low dimensional sub-manifold of the input space (Tenenbaum et al., 2000; Belkin & Niyogi, 2003). Recently, Nakada & Imaizumi (2020); Schmidt-Hieber (2019); Chen et al. (2019); Chen et al. (2019) have shown that deep ReLU network has adaptivity to the intrinsic dimensionality of data and can avoid curse of dimensionality if the intrinsic dimensionality is small. However, one drawback is that they assumed *exact* low dimensionality of the input data. This could be a strong assumption because practically observed data are always noisy, and injecting noise immediately destroys the low-dimensional structure. Therefore, we consider another direction in this paper. In terms of curse of dimensionality, Suzuki (2019) showed that deep learning can alleviate the curse of dimensionality to estimate functions in a so called mixed smooth Besov space (m-Besov). However, m-Besov space assumes strong smoothness toward *all* directions uniformly and does not include the ordinary Besov space as a special case. Moreover, the convergence rate includes heavy poly-log term which is not negligible (see Table 1).

In practice, one of the typically expected properties of a true function on high-dimensional data is that it is invariant against perturbations of an input in some specific directions (Figure 1). For example, in image-recognition tasks, the target function must be invariant against the spatial shift of an input image, which is utilized by data-augmentation techniques (Simard et al., 2003; Krizhevsky et al., 2012). In this paper, we investigate the approximation and estimation abilities of deep learning on *anisotropic Besov spaces* (Nikol'skii, 1975; Vybiral, 2006; Triebel, 2011) (also called dominated mixed-smooth Besov spaces). An anisotropic Besov space is a set of functions that have "direction-dependent" smoothness, whereas ordinary function spaces such as Hölder, Sobolev, and Besov spaces assume isotropic smoothness that is uniform in all directions. We consider a composition of functions included in an anisotropic Besov space, including several existing settings as special cases; it includes analyses of the Hölder space Schmidt-Hieber (2020) and Besov space Suzuki (2019), as well as the low-dimensional sub-manifold setting (Nakada & Imaizumi, 2020; Schmidt-Hieber, 2019; Chen et al., 2019; Chen et al., 2019)[1]. By considering such a space, we can show that deep learning can alleviate curse of dimensionality if the smoothness in each direction is highly anisotropic. Interestingly, any linear estimator (including kernel ridge regression) has worse dependence on the dimensionality than deep learning. Our contributions can be summarized as follows:

- We consider a situation in which the target function is included in a class of anisotropic Besov spaces and show that deep learning can avoid the curse of dimensionality *even if the input data*

---

[1]We would like to remark that the analysis of Nakada & Imaizumi (2020) does not require smoothness of the embedded manifold that is not covered in this paper.

*do not lie on a low-dimensional manifold.* Moreover, deep learning can achieve the optimal adaptive approximation error rate and minimax optimal estimation error rate.

- We compare deep learning with general linear estimators (including kernel methods) and show that deep learning has better dependence on the input dimensionality than linear estimators.

## 2 Problem setting and the model

In this section, we describe the problem setting considered in this work. We consider the following nonparametric regression model:

$$y_i = f^{\mathrm{o}}(x_i) + \xi_i \quad (i = 1, \ldots, n),$$

where $x_i$ is generated from a probability distribution $P_X$ on $[0,1]^d$, $\xi_i \sim N(0, \sigma^2)$, and the data $D_n = (x_i, y_i)_{i=1}^n$ are independently identically distributed. $f^{\mathrm{o}}$ is the true function that we want to estimate. We are interested in the mean squared estimation error of an estimator $\widehat{f}$: $\mathrm{E}_{D_n}[\|\widehat{f} - f^{\mathrm{o}}\|_{L^2(P_X)}^2]$, where $\mathrm{E}_{D_n}[\cdot]$ indicates the expectation with respect to the training data $D_n$. We consider a least-squares estimator in the deep neural network model as $\widehat{f}$ (see Eq. (5)) and discuss its optimality. More specifically, we investigate how the "intrinsic dimensionality" of data affects the estimation accuracy of deep learning. For this purpose, we consider an *anisotropic Besov space* as a model of the target function.

### 2.1 Anisotropic Besov space

In this section, we introduce the anisotropic Besov which was investigated as the model of the true function in this paper. Throughout this paper, we set the domain of the input to $\Omega = [0,1]^d$. For a function $f : \Omega \to \mathbb{R}$, let $\|f\|_p := \|f\|_{L^p(\Omega)} := (\int_\Omega |f|^p \mathrm{d}x)^{1/p}$ for $0 < p < \infty$. For $p = \infty$, we define $\|f\|_\infty := \|f\|_{L^\infty(\Omega)} := \sup_{x \in \Omega} |f(x)|$. For $\beta \in \mathbb{R}_{++}^d$, let $|\beta| = \sum_{j=1}^d |\beta_j|^2$.

For a function $f : \mathbb{R}^d \to \mathbb{R}$, we define the $r$th difference of $f$ in the direction $h \in \mathbb{R}^d$ as

$$\Delta_h^r(f)(x) := \Delta_h^{r-1}(f)(x+h) - \Delta_h^{r-1}(f)(x), \ \ \Delta_h^0(f)(x) := f(x),$$

for $x \in \Omega$ with $x + rh \in \Omega$, otherwise, let $\Delta_h^r(f)(x) = 0$.

**Definition 1.** *For a function $f \in L^p(\Omega)$ where $p \in (0, \infty]$, the $r$-th modulus of smoothness of $f$ is defined by $w_{r,p}(f, t) = \sup_{h \in \mathbb{R}^d : |h_i| \leq t_i} \|\Delta_h^r(f)\|_p$, for $t = (t_1, \ldots, t_d)$, $t_i > 0$.*

With this modulus of smoothness, we define the anisotropic Besov space $B_{p,q}^\beta(\Omega)$ for $\beta = (\beta_1, \ldots, \beta_d)^\top \in \mathbb{R}_{++}^d$ as follows.

**Definition 2** (Anisotropic Besov space ($B_{p,q}^\beta(\Omega)$)). *For $0 < p, q \leq \infty$, $\beta = (\beta_1, \ldots, \beta_d)^\top \in \mathbb{R}_{++}^d$, $r := \max_i \lfloor \beta_i \rfloor + 1$, let the seminorm $|\cdot|_{B_{p,q}^\alpha}$ be*

$$|f|_{B_{p,q}^\beta} := \begin{cases} \left( \sum_{k=0}^\infty [2^k w_{r,p}(f, (2^{-k/\beta_1}, \ldots, 2^{-k/\beta_d}))]^q \right)^{1/q} & (q < \infty), \\ \sup_{k \geq 0} 2^k w_{r,p}(f, (2^{-k/\beta_1}, \ldots, 2^{-k/\beta_d})) & (q = \infty). \end{cases}$$

*The norm of the anisotropic Besov space $B_{p,q}^\beta(\Omega)$ is defined by $\|f\|_{B_{p,q}^\beta} := \|f\|_p + |f|_{B_{p,q}^\beta}$, and $B_{p,q}^\beta(\Omega) = \{ f \in L^p(\Omega) \mid \|f\|_{B_{p,q}^\beta} < \infty \}$.*

Roughly speaking $\beta$ represents the smoothness in each direction. If $\beta_i$ is large, then a function in $B_{p,q}^\beta$ is smooth to the $i$th coordinate direction, otherwise, it is non-smooth to that direction. $p$ is also an important quantity that controls the *spatial inhomogeneity* of the smoothness. If $\beta_1 = \beta_2 = \cdots = \beta_d$, then the definition is equivalent to the usual Besov space (DeVore & Popov, 1988; DeVore et al., 1993). Suzuki (2019) analyzed curse of dimensionality of deep learning through a so-called *mixed smooth Besov* (m-Besov) space which imposes a stronger condition toward all directions uniformly.

---

[2]We let $\mathbb{N} := \{1, 2, 3, \ldots\}$, $\mathbb{Z}_+ := \{0, 1, 2, 3, \ldots\}$, $\mathbb{Z}_+^d := \{(z_1, \ldots, z_d) \mid z_i \in \mathbb{Z}_+\}$, $\mathbb{R}_+ := \{x \geq 0 \mid x \in \mathbb{R}\}$, and $\mathbb{R}_{++} := \{x > 0 \mid x \in \mathbb{R}\}$. We let $[N] := \{1, \ldots, N\}$ for $N \in \mathbb{N}$.

Particularly, it imposes stronger smoothness toward non-coordinate axis directions. Moreover, m-Besov space does *not* include the vanilla Besov space as a special case and thus cannot capture the situation that we consider in this paper.

Throughout this paper, for given $\beta = (\beta_1, \ldots, \beta_d)^\top \in \mathbb{R}_{++}^d$, we write $\underline{\beta} := \min_i \beta_i$ (smallest smoothness) and $\overline{\beta} := \max_i \beta_i$ (largest smoothness). The approximation error of a function in anisotropic Besov spaces is characterized by the harmonic mean of $(\beta_j)_{j=1}^d$, which corresponds to the average smoothness, and thus we define

$$\widetilde{\beta} := \left( \sum_{j=1}^d 1/\beta_j \right)^{-1}. \tag{1}$$

The Besov space is closely related to other function spaces such as Hölder space. Let $\partial^\alpha f(x) = \frac{\partial^{|\alpha|} f}{\partial^{\alpha_1} x_1 \ldots \partial^{\alpha_d} x_d}(x)$.

**Definition 3** (Hölder space ($\mathcal{C}^\beta(\Omega)$)). *For a smoothness paraemter $\beta \in \mathbb{R}_{++}$ with $\beta \notin \mathbb{N}$, consider an $m$ times differentiable function $f : \mathbb{R}^d \to \mathbb{R}$ where $m = \lfloor \beta \rfloor$ (the largest integer less than $\beta$), and let the norm of the Hölder space $\mathcal{C}^\beta(\Omega)$ be $\|f\|_{\mathcal{C}^\beta} := \max_{|\alpha| \leq m} \|\partial^\alpha f\|_\infty + \max_{|\alpha|=m} \sup_{x,y \in \Omega} \frac{|\partial^\alpha f(x) - \partial^\alpha f(y)|}{\|x-y\|^{\beta-m}}$. Then, ($\beta$-)Hölder space $\mathcal{C}^\beta(\Omega)$ is defined as $\mathcal{C}^\beta(\Omega) = \{f \mid \|f\|_{\mathcal{C}^\beta} < \infty\}$.*

Let $\mathcal{C}^0(\Omega)$ be the set of continuous functions equipped with $L^\infty$-norm: $\mathcal{C}^0(\Omega) := \{f : \Omega \to \mathbb{R} \mid f \text{ is continuous and } \|f\|_\infty < \infty\}$. These function spaces are closely related to each other.

**Proposition 1** (Triebel (2011)). *There exist the following relations between the spaces:*

1. *For $\beta = (\beta_0, \ldots, \beta_0)^\top \in \mathbb{R}^d$ with $\beta_0 \notin \mathbb{N}$, it holds that $\mathcal{C}^{\beta_0}(\Omega) = B_{\infty,\infty}^\beta(\Omega)$.*
2. *For $0 < p_1, p_2, q \leq \infty$, $p_1 \leq p_2$ and $\beta \in \mathbb{R}_{++}^d$ with $\widetilde{\beta} > (1/p_1 - 1/p_2)_+$[3], it holds that[4] $B_{p_1,q}^\beta(\Omega) \hookrightarrow B_{p_2,q}^{\gamma\beta}(\Omega)$ for $\gamma = 1 - (1/p_1 - 1/p_2)_+/\widetilde{\beta}$.*
3. *For $0 < p, q_1, q_2 \leq \infty$, $q_1 < q_2$, and $\beta \in \mathbb{R}_{++}^d$, it holds that $B_{p,q_1}^\beta \hookrightarrow B_{p,q_2}^\beta$. In particular, with properties 1 and 2, if $\widetilde{\beta} > 1/p$, it holds that $B_{p,q}^\beta(\Omega) \hookrightarrow \mathcal{C}^{\gamma\underline{\beta}}(\Omega)$ where $\gamma = 1 - 1/(\widetilde{\beta}p)$.*
4. *For $0 < p, q \leq \infty$ and $\beta \in \mathbb{R}_{++}^d$, if $\widetilde{\beta} > 1/p$, then $B_{p,q}^\beta(\Omega) \hookrightarrow \mathcal{C}^0(\Omega)$.*

This result is basically proven by Triebel (2011). For completeness, we provide its derivation in Appendix D. If the average smoothness $\widetilde{\beta}$ is sufficiently large ($\widetilde{\beta} > 1/p$), then the functions in $B_{p,q}^\beta$ are continuous; however, if it is small ($\widetilde{\beta} < 1/p$), then they are no longer continuous. Small $p$ indicates spatially inhomogeneous smoothness; thus, spikes and jumps appear (see Donoho & Johnstone (1998) for this perspective, from the viewpoint of wavelet analysis).

## 2.2 Model of the true function

As a model of the true function $f^\circ$, we consider two types of models: *Affien composition model* and *deep composition model*. For a Banach space $\mathcal{H}$, we let $U(\mathcal{H})$ be the unit ball of $\mathcal{H}$.

**(a) Affine composition model:** The first model we introduced is a very naive model which is just a composition of an affine transformation and a function in the anisotropic Besov space:

$$\mathcal{H}_{\text{aff}} := \{h(Ax + b) \mid h \in U(B_{p,q}^\beta([0,1]^{\tilde{d}})), A \in \mathbb{R}^{\tilde{d} \times d}, b \in \mathbb{R}^b \text{ s.t. } Ax + b \in [0,1]^{\tilde{d}} \ (\forall x \in \Omega)\},$$

where we assume $\tilde{d} \leq d$. Here, we assumed that the affine transformation has an appropriate scaling such that $Ax + b$ is included in the domain of $h$ for all $x \in \Omega$. This is a quite naive model but provides an instructive example to understand how the estimation error of deep learning behaves under the anisotropic setting.

**(b) Deep composition model:** The *deep composition model* generalizes the affine composition model to a composition of nonlinear functions. Let $m_1 = d$, $m_{L+1} = 1$, $m_\ell$ be the dimension of the

---

[3]Here, we let $(x)_+ := \max\{x, 0\}$.

[4]The symbol $\hookrightarrow$ means continuous embedding.

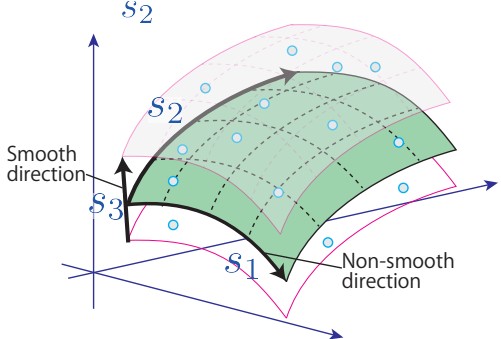

Figure 1: Near low dimensional data distribution with anisotropic smoothness of the target function. The target function has less smoothness $(s_1, s_2)$ toward the first two coordinates on the manifold while it is almost constant toward the third coordinate (large $s_3$).

$\ell$th layer, and let $\beta^{(\ell)} \in \mathbb{R}^{m_\ell}_{++}$ be the smoothness parameter in the $\ell$th layer. The deep composition model is defined as

$$\mathcal{H}_{\mathrm{deep}} := \{h_H \circ \cdots \circ h_1(x) \mid h_\ell : [0,1]^{m_\ell} \to [0,1]^{m_{\ell+1}}, \; h_{\ell,k} \in U(B_{p,q}^{\beta^{(\ell)}}([0,1]^{m_\ell})) \; (\forall k \in [m_{\ell+1}])\}.$$

Here, the interval $[0,1]$ can be replaced by another compact interval, such as $[a_\ell, b_\ell]$, but this difference can be absorbed by changing a scaling factor. The assumption $\|h_{\ell,k}\|_{B_{p,q}^{\beta^{(\ell)}}} \le 1$ can also be relaxed, but we do not pursue that direction due to presentation simplicity. This model includes the affine composition model as a special case. However, it requires a stronger assumption to properly evaluate the estimation error on this model.

**Examples**  The model we have introduced includes some instructive examples as listed below:

**(a) Linear projection**  Schmidt-Hieber (2020) analyzed estimation of the following model by deep learning: $f^{\mathrm{o}}(x) = g(w^\top x)$ where $g \in \mathcal{C}^\beta([0,1])$ and $w \in \mathbb{R}^d$. In this example, the function $f^{\mathrm{o}}$ varies along only one direction, $w$. Apparently, this is an example of the affine composition model.

**(b) Distribution on low dimensional smooth manifold**  Assume that the input $x$ is distributed on a low-dimensional smooth manifold embedded in $\Omega$, and the smoothness of the true function $f^{\mathrm{o}}$ is anisotropic along a coordinate direction on the manifold. We suppose that the low dimensional manifold is $\tilde{d}$-dimensional and $\tilde{d} \ll d$. In this situation, the true function can be written as $f^{\mathrm{o}}(x) = h(\phi(x))$ where $\phi : \mathbb{R}^d \to \mathbb{R}^{\tilde{d}}$ is a map that returns the coordinate of $x$ on the manifold and $h$ is an element in an anisotropic Besov space on $\mathbb{R}^{\tilde{d}}$. This situation appears if data is distributed on a low-dimensional sub-manifold of $\Omega$ and the target function is invariant against noise injection to some direction on the manifold at each input point $x$ (Figure 1 illustrates this situation). One typical example of this situation is a function invariant with data augmentation (Simard et al., 2003; Krizhevsky et al., 2012). Even if the noise injection destroys low dimensionality of the data distribution (i.e., $\tilde{d} = d$), an anisotropic smoothness of the target function eases the curse of dimensionality as analyzed below, which is quite different from existing works (Yang & Dunson, 2016; Bickel & Li, 2007; Nakada & Imaizumi, 2020; Schmidt-Hieber, 2019; Chen et al., 2019; Chen et al., 2019).

**Related work**  Here, we introduce some more related work and discuss their relation to our analysis. The statistical analysis on an anisotropic Besov space can be back to Ibragimov & Khas'minskii (1984) who considered density estimation, where the density is assumed to be included in an anisotropic Sobolev space with $p \ge 2$, and derived the minimax optimal rate $n^{-r\widetilde{\beta}/(2\widetilde{\beta}+1)}$ with respect to $L^r$-norm. Nyssbaum (1983, 1987) analyzed a nonparametric regression problem on an anisotropic Besov space. Following these results, several studied have been conducted in the literature pertaining to nonparametric statistics, such as nonlinear kernel estimator Kerkyacharian et al. (2001), adaptive confidence band construction Hoffman & Lepski (2002), optimal aggregation Gaiffas & Lecue (2011), Gaussian process estimator Bhattacharya et al. (2011, 2014), and kernel ridge regression Hang & Steinwart (2018). Basically, these studies investigated estimation problems in which the target function is in anisotropic Besov spaces, but the composition models considered in this paper have not been analyzed. Hoffman & Lepski (2002); Bhattacharya et al. (2011) considered a dimension reduction model; that is, the target function is dependent on only a few variables of $x$, but they did not deal with more general models, such as the affine/deep composition models. The nonparametric regression problems where the input data are distributed on a low-dimensional smooth manifold has been studied as a "manifold regression" Yang & Dunson (2016); Bickel & Li

(2007); Yang & Tokdar (2015). Such a model can be considered as a specific example of the deep composition model. In this sense, our analysis is a significant extension of these analyses.

# 3 Approximation error analysis

Here, we consider approximating the true function $f^o$ via a deep neural network and derive the approximation error. As the activation function, we consider the ReLU activation denoted by $\eta(x) = \max\{x, 0\}$ ($x \in \mathbb{R}$). For a vector $x$, $\eta(x)$ is operated in an element-wise manner. The model of neural networks with height $L$, width $W$, sparsity constraint $S$, and norm constraint $B$ as $\Phi(L, W, S, B) := \{(\mathcal{W}^{(L)}\eta(\cdot) + b^{(L)}) \circ \cdots \circ (\mathcal{W}^{(1)}x + b^{(1)}) \mid \mathcal{W}^{(L)} \in \mathbb{R}^{1 \times W}, b^{(L)} \in \mathbb{R}, \mathcal{W}^{(1)} \in \mathbb{R}^{W \times d}, b^{(1)} \in \mathbb{R}^W, \mathcal{W}^{(\ell)} \in \mathbb{R}^{W \times W}, b^{(\ell)} \in \mathbb{R}^W (1 < \ell < L), \sum_{\ell=1}^{L}(\|\mathcal{W}^{(\ell)}\|_0 + \|b^{(\ell)}\|_0) \le S, \max_\ell \|\mathcal{W}^{(\ell)}\|_\infty \vee \|b^{(\ell)}\|_\infty \le B\}$, where $\| \cdot \|_0$ is the $\ell_0$-norm of the matrix (the number of non-zero elements of the matrix), and $\| \cdot \|_\infty$ is the $\ell_\infty$-norm of the matrix (maximum of the absolute values of the elements). The sparsity constraint and norm bounds are required to obtain the near-optimal rate of the estimation error. To evaluate the accuracy of the deep neural network model in approximating target functions, we define the worst-case approximation error as

$$R_r(\mathcal{F}, \mathcal{H}) := \sup_{f^* \in \mathcal{H}} \inf_{f \in \mathcal{F}} \|f^* - f\|_{L^r(\Omega)},$$

where $\mathcal{F}$ is the set of functions used for approximation, and $\mathcal{H}$ is the set of target functions.

**Proposition 2** (Approximation ability for anisotropic Besov space). *Suppose that $0 < p, q, r \le \infty$ and $\beta \in \mathbb{R}^d_{++}$ satisfy the following condition: $\widetilde{\beta} > (1/p - 1/r)_+$. Assume that $m \in \mathbb{N}$ satisfies $0 < \overline{\beta} < \min(m, m - 1 + 1/p)$. Let $\delta = (1/p - 1/r)_+$, $\nu = (\widetilde{\beta} - \delta)/(2\delta)$ and $W_0(d) := 6dm(m + 2) + 2d$. Then, for $N \in \mathbb{N}$, we can bound the approximation error as*

$$R_r(\Phi(L_1, W_1, S_1, B_1), U(B_{p,q}^\beta(\Omega))) \lesssim N^{-\widetilde{\beta}},$$

*by setting*

$$L_1(d) := 3 + 2\lceil \log_2 \left( \frac{3^{d \vee m}}{\epsilon c_{(d,m)}} \right) + 5\rceil \lceil \log_2(d \vee m) \rceil, \quad W_1(d) := NW_0, \tag{2}$$

$$S_1(d) := [(L - 1)W_0^2 + 1]N, \quad B_1(d) := O(N^{d(1+\nu^{-1})(1/p - \widetilde{\beta})_+}), \tag{3}$$

*for $\epsilon = N^{-\widetilde{\beta}} \log(N)^{-1}$ and a constant $c_{(d,m)}$ depending only on $d$ and $m$.*

The proof of this proposition is provided in Appendix B. The rate $N^{-\widetilde{\beta}}$ is the optimal *adaptive* approximation error rate that can be achieved by a model with $N$ parameters (the difference between adaptive and non-adaptive methods is explained in the discussion below). Note that this is an approximation error in an oracle setting and no sample complexity appears here. We notice that we can avoid the *curse of dimensionality* if the average smoothness $\widetilde{\beta}$ is small. This means that if the target function is non-smooth in only a few directions and smooth in other directions, we can avoid the curse of dimensionality. In contrast, if we consider an isotropic Besov space where $\beta_1 = \cdots = \beta_d(= \underline{\beta})$, then $\widetilde{\beta} = \underline{\beta}/d$, which directly depends on the dimensionality $d$, and we need an exponentially large number of parameters in this situation to achieve $\epsilon$-accuracy. Therefore, the anisotropic smoothness has a significant impact on the approximation error rate. The assumption $\widetilde{\beta} > (1/p - 1/r)_+$ ensures the $L_r$-integrability of the target function, and the inequality (without equality) admits a near-optimal wavelet approximation of the target function in terms of $L_r$-norm.

Using this evaluation as a basic tool, we can obtain the approximation error for the deep composition models. We can also obtain the approximation error for the affine composition models, but it is almost identical to Proposition 2. Therefore, we defer it to Appendix A.

**Theorem 1** (Deep composition model). *Assume that $\widetilde{\beta}^{(\ell)} > 1/p$ for all $\ell = 1, \ldots, H$. Then, the estimation error on the deep composition model is bounded as*

$$R_\infty(\Phi(L, W, S, B), \mathcal{H}_{\text{deep}}) \lesssim \max_{\ell \in [H]} N^{-\widetilde{\beta}^{*(\ell)}}, \tag{4}$$

*where $\widetilde{\beta}^{*(\ell)} = \widetilde{\beta}^{(\ell)} \prod_{k=\ell+1}^{H}[(\underline{\beta}^{(k)} - 1/p) \wedge 1]$, and $L = \sum_{\ell=1}^{H}(L_1(m_\ell) + 1), W = \max_\ell(W_1(m_\ell) \vee m_{\ell+1}), S = \sum_{\ell=1}^{H}(S_1(m_\ell) + 3m_{\ell+1}), B = \max_\ell B_1(m_\ell)$.*

The proof can be found in Appendix B.1. Since the model is more general than the vanilla anisotropic Besov space, we require a stronger assumption $\widetilde{\beta}^{(\ell)} > 1/p$ on $\widetilde{\beta}^{(\ell)}$ than the condition in Proposition 2. This is because we need to bound the Hölder smoothness of the remaining layers to bound the influence of the approximation error in the internal layers to the entire function. Hölder smoothness is ensured according to the embedding property under this condition (Proposition 1). This Hölder smoothness assumption affects the approximation error rate. The convergence rate $\widetilde{\beta}^{*(\ell)}$ in Eq. (4) is different from that in Eq. (8). This is because the approximation error in the internal layers are propagated through the remaining layers with Hölder smoothness and its amplitude is controlled by the Hölder smoothness.

**Approximation error by non-adaptive method**  The approximation error obtained in the previous section is called an adaptive error rate in the literature regarding approximation theory (DeVore, 1998). If we fix $N$ bases beforehand and approximate the target function by a linear combination of the $N$ bases (which is called the non-adaptive method), then we *cannot* achieve the adaptive error rate obtained in the previous section. Roughly speaking, the approximation error of non-adaptive methods is lower bounded by $N^{-\left(\widetilde{\beta}-\left(\frac{1}{p}-\frac{1}{\min\{2,r\}}\right)_+\right)}$ (Myronyuk, 2015, 2016, 2017), which is slower than the approximation error rate of deep neural networks especially for small $p$.

## 4 Estimation error analysis

In this section, we analyze the accuracy of deep learning in estimating a function in compositions of anisotropic Besov spaces. We consider a least-squares estimator in the deep neural network model:

$$\widehat{f} = \operatorname{argmin}_{\bar{f}: f \in \Phi(L, W, S, B)} \sum_{i=1}^{n} (y_i - \bar{f}(x_i))^2 \tag{5}$$

where $\bar{f}$ is the *clipping* of $f$ defined by $\bar{f} = \min\{\max\{f, -F\}, F\}$ for a constant $F > 0$ which is realized by ReLU units. The network parameters $(L, W, S, B)$ should be specified appropriately as indicated in Theorems 2 and 3. In practice, these parameters can be specified by cross validation. Indeed, we can theoretically show that cross validation can provide the appropriate choice of these parameters in compensation to an additional $\log(n)$-factor in the estimation error bound. This estimator can be seen as a sparsely regularized estimator because there are constraints on $S$. In terms of optimization, this requires a combinatorial optimization, but we do not pursue the computational aspect. The estimation error that we derive in this section can involve the optimization error, but for simplicity, we only demonstrate the estimation error of the *ideal* situation where there is no optimization error.

**Affine composition model**  The following theorem provides an upper bound of the estimation error for the affine composition model.

**Theorem 2.** *Assume the same condition as in Theorem 6; in particular, suppose $0 < p, q \leq \infty$ and $\widetilde{\beta} > (1/p - 1/2)_+$ for $\widetilde{\beta} \in \mathbb{R}_{++}^{\tilde{d}}$. Moreover, we assume that the distribution $P_X$ has a density $p_X$ such that $\|p_X\|_\infty \leq R$ for a constant $R > 0$. If $f^\circ \in \mathcal{H}_{\mathrm{aff}} \cap L^\infty(\Omega)$, and $\|f^\circ\|_\infty \leq F$ for $F \geq 1$; then, letting $(W, L, S, B) = (L_1(\tilde{d}), W_1(\tilde{d}), S_1(\tilde{d}), (\tilde{d}C + 1)B_1(\tilde{d}))$ as in Theorem 6 with $N \asymp n^{\frac{1}{2\tilde{\beta}+1}}$, we obtain*

$$\mathrm{E}_{D_n}[\|f^\circ - \widehat{f}\|_{L^2(P_X)}^2] \lesssim n^{-\frac{2\widetilde{\beta}}{2\widetilde{\beta}+1}} \log(n)^3,$$

*where $\mathrm{E}_{D_n}[\cdot]$ indicates the expectation with respect to the training data $D_n$.*

The proof is provided in Appendix C. We will show that the convergence rate $n^{-2\widetilde{\beta}/(2\widetilde{\beta}+1)}$ is minimax optimal in Section 5 (see also Kerkyacharian & Picard (1992); Donoho et al. (1996); Donoho & Johnstone (1998); Giné & Nickl (2015) for ordinary Besov spaces). The $L^\infty$-norm constraint $\|f^\circ\|_\infty \leq F$ is used to derive a uniform bound on the discrepancy between the population and the empirical $L^2$-norm. Without this condition, the convergence rate could be slower.

**Deep composition model**  For the deep composition model, we obtain the following convergence rate. This is an extension of Theorem 2 but requires a stronger assumption on the smoothness.

**Theorem 3.** *Suppose that $0 < p, q \leq \infty$ and $\widetilde{\beta}^{(\ell)} > 1/p$ for all $\ell \in [H]$. If $f^\circ \in \mathcal{H}_{\mathrm{deep}} \cap L^\infty(\Omega)$, and $\|f\|_\infty \leq F$ for $F \geq 1$, then we obtain*

$$\mathrm{E}_{D_n}[\|f^\circ - \widehat{f}\|_{L^2(P_X)}^2] \lesssim \max_{\ell \in [H]} n^{-2\widetilde{\beta}^{*(\ell)}/(2\widetilde{\beta}^{*(\ell)}+1)} \log(n)^3,$$

where $\widetilde{\beta}^{*(\ell)}$ is defined in Theorem 1, and $(L, W, S, B)$ is as given in Theorem 1 with $N \asymp \max_{\ell \in [L]} n^{\frac{1}{2\widetilde{\beta}^{*(\ell)}+1}}$.

The proof is provided in Appendix C. We will show that this is also minimax optimal in Theorem 4. Because of the Hölder continuity, the convergence rate becomes slower than the affine composition model (that is, $\widetilde{\beta}^{*(\ell)} \leq \widetilde{\beta}^{(\ell)}$). However, this slower rate is unavoidable in terms of the minimax optimal rate. Schmidt-Hieber (2020) analyzed the same situation for the Hölder class which corresponds to $\beta_1^{(\ell)} = \cdots = \beta_d^{(\ell)}$ ($\forall \ell$) and $p = q = \infty$. Our analysis far extends their analysis to the setting of anisotropic Besov spaces in which the parameters $\beta^{(\ell)}, p, q$ have much more freedom.

From these two bounds (Theorems 2 and 3), we can see that as the smoothness $\widetilde{\beta}$ becomes large, the convergence rates faster. If the target function is included in the isotropic Besov space with smoothness $\beta_1 = \cdots = \beta_d(= \underline{\beta})$, then the estimation error becomes

**(Isotropic Besov)** $\qquad\qquad\qquad n^{-2\underline{\beta}/(2\underline{\beta}+d)}.$

In the exponent, the dimensionality $d$ appears, which causes the curse of dimensionality. In contrast, if the target function is in the anisotropic Besov space, and the smoothness in each direction is sufficiently imbalanced such that $\widetilde{\beta}$ does not depend on $d$, our obtained rate

**(Anisotropic Besov)** $\qquad\qquad\qquad n^{-2\widetilde{\beta}/(2\widetilde{\beta}+1)}$

avoids the curse of dimensionality. For high-dimensional settings, there would be several redundant directions in which the true function does not change. Deep learning is adaptive to this redundancy and achieves a better estimation error. However, in Section 6, we prove that linear estimators are affected by the dimensionality more strongly than deep learning. This indicates the superiority of deep learning.

## 5 Minimax optimal rate

Here, we show that the estimation error rate, that we have presented, of deep learning achieves the *minimax optimal rate*. Roughly speaking the minimax optimal risk on a model $\mathcal{F}^\circ$ of the true function is the smallest worst case error over all estimators:

$$R_*(\mathcal{F}^\circ) := \inf_{\widehat{f}} \sup_{f^\circ \in \mathcal{F}^\circ} \mathrm{E}_{D_n}[\|\widehat{f} - f^\circ\|_{L^2(P_X)}^2],$$

where $\widehat{f}$ runs over all estimators. The convergence rate of the minimax optimal risk is referred to as minimax optimal rate. We obtain the following minimax optimal rate for anisotropic Besov spaces.

**Theorem 4. (a) Affine composition model:** *For $0 < p, q \leq \infty$ and $\beta \in \mathbb{R}_{++}^d$, assume that $\widetilde{\beta} > \max\{1/p - 1/2, 1/p - 1, 0\}$. Then, the minimax optimal risk of the affine composition model is lower bounded as $R_*(\mathcal{H}_{\mathrm{aff}}) \gtrsim n^{-\frac{2\widetilde{\beta}}{2\widetilde{\beta}+1}}$.* **(b) Deep composition model:** *For $0 < p, q \leq \infty$ and $\beta^{(\ell)} \in \mathbb{R}_{++}^d$ ($\ell = 1, \dots, H$), assume that $\widetilde{\beta}^{(\ell)} > 1/p$. Let $\epsilon > 0$ be arbitrarily small for $q < \infty$, and let $\epsilon = 0$ for $q = 0$. Let $\widetilde{\beta}^{*(\ell)} = \widetilde{\beta}^{(\ell)} \prod_{k=\ell+1}^H [(\underline{\beta}^{(k)} - 1/p + \epsilon) \wedge 1]$, and $\widetilde{\beta}^{**} := \min_\ell \widetilde{\beta}^{*(\ell)}$. Then, the minimax optimal risk of the deep composition model is lower bounded as $R_*(\mathcal{H}_{\mathrm{deep}}) \gtrsim n^{-\frac{2\widetilde{\beta}^{**}}{2\widetilde{\beta}^{**}+1}}$.*

The proof is provided in Appendix E (see also Ibragimov & Khas'minskii (1984); Nyssbaum (1987)). From this theorem, we can see that the estimation error of deep learning shown in Theorems 2 and 3 indeed achieve the minimax optimal rate up to a poly-$\log(n)$ factor.

## 6 Suboptimality of linear estimators

In this section, we give the minimax optimal rate in the class of *linear estimators*. The linear estimator is a class of estimators that can be written as

$$\widehat{f}(x) = \sum_{i=1}^n y_i \varphi_i(x; X^n),$$

where $X^n = (x_1, \dots, x_n)$ and $\varphi_i(x; X^n)$ ($i = 1, \dots, n$) are (measurable) functions that only depend on $x$ and $X^n$. This is linearly dependent on $Y^n = (y_1, \dots, y_n)$. We notice that the kernel

ridge regression is included in this class because it can be written as $\widehat{f}(x) = k_{x,X^n}(k_{X^n,X^n} + \lambda\mathrm{I})^{-1}Y^n$, which linearly depends on $Y^n$. This class includes other important estimators, such as the Nadaraya–Watson estimator, the $k$-nearest neighbor estimator, and the sieve estimator. We compare deep learning with the linear estimators in terms of minimax risk. For this purpose, we define the minimax risk of the class of linear estimators:

$$R_*^{(\mathrm{lin})}(\mathcal{F}^\circ) := \inf_{\widehat{f}:\text{ linear}} \sup_{f^\circ \in \mathcal{F}^\circ} \mathrm{E}_{D_n}[\|\|f^\circ - \widehat{f}\|_{L^2(P_X)}^2],$$

where $\widehat{f}$ runs over all *linear estimators*. We can see that linear estimators suffer from the sub-optimal rate because of the following two points: (i) they do not have adaptivity, and (ii) they significantly suffer from the curse of dimensionality.

**Theorem 5.** *(i) Suppose that the input distribution $P_X$ is the uniform distribution on $\Omega = [0,1]^d$ and assume that $\widetilde{\beta} > 1/p$ and $1 \le p, q \le \infty$. Then, the minimax optimal rate of the linear estimators is lower bounded as*

$$R_*^{(\mathrm{lin})}(U(B_{p,q}^\beta)) \gtrsim n^{-\frac{2\widetilde{\beta}-v}{2\widetilde{\beta}+1-v}}, \tag{6}$$

*where $v = 2(1/p - 1/2)_+$.*

*(ii) In addition to the above conditions, we assume that the true function is included in the affine composition model with $\tilde{d} \le d$, $\underline{\beta} = \beta_1 = \cdots = \beta_{\tilde{d}}$ and $0 < p \le 2$. Let $a_d = 1 + \kappa$ with arbitrary small $\kappa > 0$ when $\tilde{d} < d/2$, and let $a_d = 0$ when $\tilde{d} \ge d/2$. Then, the minimax rate of the linear estimators on the affine composition model is lower bounded by*

$$R_*^{(\mathrm{lin})}(\mathcal{H}_{\mathrm{aff}}) \gtrsim n^{-\frac{2(\underline{\beta}-\tilde{d}/p+d/2+a_d)}{2(\underline{\beta}-\tilde{d}/p+d/2+a_d)+d}}. \tag{7}$$

The proof is provided in Appendix F. (i) The lower bound (7) reveals the suboptimality of linear estimators in terms of input dimensionality. Actually, if we consider a particular case where $\tilde{d} = 1$, $p = 1$ and $d \gg \tilde{d}$, then the obtained minimax rate of linear estimators and the estimation error rate of deep learning can be summarized as

$$\text{linear}: \ n^{-\frac{2\underline{\beta}+d}{2\underline{\beta}+2d}}, \qquad \text{deep}: \ n^{-\frac{2\underline{\beta}}{2\underline{\beta}+1}},$$

by Theorem 2 when $\underline{\beta} > 1$ (which can be checked by noticing $\tilde{d} = \underline{\beta}/\widetilde{\beta} = 1$ in this situation). We can see that the dependence on the dimensionality of linear estimators is significantly worse than that of deep leaning. This indicates poor adaptivity of linear estimators to the intrinsic dimensionality of data. Actually, as $d$ becomes large, the rate for the linear estimator approaches to $1/\sqrt{n}$ but that for the deep learning is not affected by $d$ and still faster than $1/\sqrt{n}$. To show the theorem, we used the "convex-hull argument" developed by Hayakawa & Suzuki (2019); Donoho & Johnstone (1998). We combined this technique with the so-called *Irie-Miyake's integral representation* (Irie & Miyake, 1988; Hornik et al., 1990). Note that this difference appears because there is an affine transformation in the first layer of the affine composition model. Deep learning is flexible against such a coordinate transform so that it can find directions to which the target function is smooth. In contrast, kernel methods do not have such adaptivity because there is no feature extraction layer. (ii) The lower bound (6) states that when $p < 2$ (that is, $v > 0$), the minimax rate of the linear estimators is outperformed by that of deep learning (Theorem 2). This is due to the "adaptivity" of deep leaning. When $p$ is small, the smoothness of the target function is less homogeneous, and it requires an adaptive approximation scheme to achieve the best estimation error. Linear estimators do not have adaptivity and thus fail to achieve the minimax optimal rate. Our bound (6) extends the result by Zhang et al. (2002) to a multivariate anisotropic Besov space while Zhang et al. (2002) investigated the univariate space ($d = 1$).

## 7 Conclusion

We investigated the approximation error and estimation error of deep learning in the anisotropic Besov spaces. It was proved that the convergence rate is determined by the average of the anisotropic smoothness, which results in milder dependence on the input dimensionality. If the smoothness is

highly anisotropic, deep learning can avoid overfitting. We also compared the error rate of deep learning with that of linear estimators and showed that deep learning has better dependence on the input dimensionality. Moreover, it was shown that deep learning can achieve the adaptive rate and outperform non-adaptive approximation methods and linear estimators if the homogeneity $p$ of smoothness is small. These analyses strongly support the practical success of deep learning from a theoretical perspective.

**Limitations of this work**  Our work does not cover the optimization aspect of deep learning. It is assumed that the regularized least squares (5) can be executed. It would be nice to combine our study with recent developments of non-convex optimization techniques (Vempala & Wibisono, 2019; Suzuki & Akiyama, 2021).

**Potential negative societal impact**  Since this is purely theoretical result, it is not expected that there is a direct negative societal impact. However, revealing detailed properties of the deep learning could promote an opportunity to pervert deep learning.

## Acknowledgment

TS was partially supported by JSPS KAKENHI (18H03201), Japan Digital Design and JST CREST. AN was partially supported by JSPS Kakenhi (19K20337) and JST-PRESTO.

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
