# ——Appendix——

## A  Approximation error of Affine composition model

**Theorem 6** (Affine composition model). *Assume that the distribution of $\tilde{x} = Ax + b \in \mathbb{R}^{\tilde{d}}$ has a bounded density function on $[0,1]^{\tilde{d}}$ when $x$ obeys the uniform distribution on $\Omega$, and each element in $A$ and $b$ is bounded by a constant $C$. Assume that $0 < p, q, r \leq \infty$ and $\beta \in \mathbb{R}^{\tilde{d}}_{++}$ satisfy $\widetilde{\beta} > (1/p - 1/r)_+$. Then, it holds that*

$$R_r(\Phi(L_1(\tilde{d}), W_1(\tilde{d}), S_1(\tilde{d}), (\tilde{d}C + 1)B_1(\tilde{d})), \mathcal{H}_{\mathrm{aff}}) \lesssim N^{-\widetilde{\beta}}, \tag{8}$$

*where $L_1(\cdot)$, $W_1(\cdot)$, $S_1(\cdot)$, $B_1(\cdot)$ are defined in Eq.* (3).

The assumption $\widetilde{\beta} > (1/p - 1/r)_+$ ensures the $L_r$-integrability of the target function, and the inequality (without equality) admits a near-optimal wavelet approximation of the target function in terms of $L_r$-norm. From this theorem, the approximation error is almost identical to that for $B^\beta_{p,q}(\Omega)$ (Proposition 2).

## B  Proofs of approximation error bounds

To show the approximation accuracy, a key step is to show that the ReLU neural network can approximate the *cardinal B-spline* with high accuracy. Let $\mathcal{N}(x) = 1$ ($x \in [0,1]$), $0$ (otherwise), then the *cardinal B-spline of order $m$* is defined by taking $m + 1$-times convolution of $\mathcal{N}$:

$$\mathcal{N}_m(x) = (\underbrace{\mathcal{N} * \mathcal{N} * \cdots * \mathcal{N}}_{m + 1 \text{ times}})(x),$$

where $f * g(x) := \int f(x - t)g(t)\mathrm{d}t$. It is known that $\mathcal{N}_m$ is a piece-wise polynomial of order $m$. For $k \in \mathbb{Z}^d_+$ and $j = (j_1, \ldots, j_d) \in \mathbb{Z}^d_+$, let

$$M^d_{k,j}(x) = \prod_{i=1}^d \mathcal{N}_m(2^{\lfloor k\beta'_i \rfloor} x_i - j_i),$$

where $\beta \in \mathbb{R}^d_{++}$ is a given smoothness parameter (we omit the dependency on $\beta$ from the notation which would be obvious from the context). Here, $k$ controls the spatial "resolution" and $j$ specifies the location on which the basis is put. Basically, we approximate a function $f$ in an anisotropic Besov space via super-position of $M^m_{k,j}(x)$, which is closely related to wavelet analysis (Mallat, 1999). The following is a key lemma that was proven by Suzuki (2019).

**Lemma 1** (Approximation of cardinal B-spline basis by the ReLU activation). *There exists a constant $c_{(d,m)}$ depending only on $d$ and $m$ such that, for all $\epsilon > 0$, there exists a neural network $\check{M} \in \Phi(L_0, W_0, S_0, B_0)$ with $L_0 := 3 + 2\left\lceil \log_2\left(\frac{3^{d\vee m}}{\epsilon c_{(d,m)}}\right) + 5\right\rceil \lceil \log_2(d \vee m) \rceil$, $W_0 := 6dm(m+2) + 2d$, $S_0 := L_0 W_0^2$ and $B_0 := 2(m+1)^m$ that satisfies*

$$\|M^d_{0,0} - \check{M}\|_{L^\infty(\mathbb{R}^d)} \leq \epsilon,$$

*and $\check{M}(x) = 0$ for all $x \notin [0, m+1]^d$.*

Let

$$\|k\|_{\underline{\beta}/\beta} := \sum_{j=1}^d \lfloor k\underline{\beta}/\beta_j \rfloor$$

for a $k \in \mathbb{Z}$. For order $m \in \mathbb{N}$ of the cardinal B-spline bases, let

$$J_i(k) = \{-m, -m+1, \ldots, 2^{\lfloor k\beta'_i \rfloor} - 1, 2^{\lfloor k\beta'_i \rfloor}\}$$

and

$$J(k) := J_1(k) \times J_2(k) \times \cdots \times J_d(k).$$

and the quasi-norm of the coefficient $(\alpha_{k,j})_{k,j}$ for $k \in \mathbb{Z}_+$ and $j \in J(k)$ be

$$\|(\alpha_{k,j})_{k,j}\|_{b_{p,q}^{\beta}} := \left\{ \sum_{k=0}^{\infty} \left[ 2^{k[\underline{\beta} - (\sum_{i=1}^{d} \lfloor k\beta_i' \rfloor / k)/p]} \left( \sum_{j \in J(k)} |\alpha_{k,j}|^p \right)^{1/p} \right]^q \right\}^{1/q}.$$

For $p = \infty$ or $q = \infty$, the definition should be appropriately modified as usual.

**Lemma 2.** *Assume the condition $\widetilde{\beta} > (1/p - 1/r)_+$ in Proposition 2 and $0 < \overline{\beta} < \min(m, m-1+ 1/p)$ where $m \in \mathbb{N}$ is the order of the cardinal B-spline bases. Then, $f \in B_{p,q}^{\beta}$ admits the following decomposition:*

$$f = \sum_{k=0}^{\infty} \sum_{j \in J(k)} \alpha_{k,j} M_{k,j}^d(x) \tag{9}$$

*with convergence in the sense of $L^p$, and the coefficient $(\alpha_{k,j})$ yields the following norm equivalence*

$$\|f\|_{B_{p,q}^{\beta}} \simeq \|(\alpha_{k,j})_{k,j}\|_{b_{p,q}^{\beta}}. \tag{10}$$

*For an integer $K \in \mathbb{N}$, let $N = \lceil 2^{\|K\|_{\underline{\beta}/\beta}} \rceil$, then for any $f \in B_{p,q}^{\beta}(\Omega)$, there exists $f_N$ that satisfies*

$$\|f - f_N\|_{L^r(\Omega)} \lesssim N^{-\widetilde{\beta}} \|f\|_{B_{p,q}^{\beta}},$$

*and has the following form:*

$$f_N(x) = \sum_{k=0}^{K} \sum_{j \in J(k)} \alpha_{k,j} M_{k,j}^d(x) + \sum_{k=K+1}^{K^*} \sum_{i=1}^{n_k} \alpha_{k,j_i} M_{k,j_i}^d(x), \tag{11}$$

*where $K^* = \lceil K(1 + 1/\nu) \rceil$, $n_k = \lceil 2^{\|K\|_{\underline{\beta}/\beta} - \epsilon(\|k\|_{\underline{\beta}/\beta} - \|K\|_{\underline{\beta}/\beta})} \rceil$ $(k = K+1, \ldots, K^*)$ for $\delta = (1/p - 1/r)_+$ and $\nu = (\widetilde{\beta} - \delta)/(2\delta)$, and $(j_i)_{i=1}^{n_k} \subset J(k)$.*

*Proof of Lemma 2.* Leisner (2003) showed that there exists a bounded linear operator $P_k$ that can be expressed as

$$P_k(f)(x) = \sum_{j \in J(k)} a_{k,j} M_{k,j}^d(x) \tag{12}$$

where $\alpha_{k,j}$ is constructed in a certain way, and for every $f \in L^p([0,1]^d)$ with $0 < p \leq \infty$, it holds

$$\|f - P_k(f)\|_{L^p} \leq C w_{r,p}(f, (2^{-k\beta_1'}, \ldots, 2^{-k\beta_d'})),$$

(See Theorem 3.2.4 of Leisner (2003) and DeVore & Popov (1988)). Let

$$p_k(f) := P_k(f) - P_{k-1}(f), \quad P_{-1}(f) = 0.$$

Then, Leisner (2003) showed that when $0 < p, q \leq \infty$ and $0 < \overline{\beta} < \min(m, m-1+1/p)$, $f$ belongs to $B_{p,q}^{\beta}$ if and only if $f$ can be decomposed into

$$f = \sum_{k=0}^{\infty} p_k(f),$$

with the convergence condition

$$\|(p_k(f))_{k=0}^{\infty}\|_{b_q^{\beta}(L^p)} := \left[ \sum_{k=0}^{\infty} (2^{\underline{\beta}k} \|p_k\|_{L^p})^q \right]^{1/q} < \infty.$$

In particular, it is shown that

$$\|f\|_{B_{p,q}^s} \simeq \|(p_k(f))_{k=0}^{\infty}\|_{b_p^s(L^p)}. \tag{13}$$

Here, each $p_k$ can be expressed as $p_k(x) = \sum_{j \in J(k)} \alpha_{k,j} M_{k,j}^d(x)$ for a coefficient $(\alpha_{k,j})_{k,j}$ which could be different from $(a_{k,j})_{k,j}$ appearing in Eq. (12), and thus $f \in B_{p,q}^\beta$ can be decomposed into

$$f = \sum_{k=0}^\infty \sum_{j \in J(k)} \alpha_{k,j} M_{k,j}^d(x)$$

with convergence in the sense of $L^p$. Moreover, it is shown that $\|p_k\|_{L^p} \simeq (2^{-kd} \sum_{j \in J(k)} |\alpha_{k,j}|^p)^{1/p}$ and thus

$$\|f\|_{B_{p,q}^\beta} \simeq \|(\alpha_{k,j})_{k,j}\|_{b_{p,q}^\beta}.$$

This yields the first assertion.

Next, we move to the second assertion. If $p \geq r$, the assertion can be shown in the same manner as Theorem 3.1 of Dũng (2011a). More precisely, we can show the assertion in a similar line to the following proof for $p < r$ by setting $K = K^*$. Thus, we show the assertion only for $p < r$. In this regime, we need to use an adaptive approximation method. In the following, we assume $p < r$. For a given $K$, by appropriately choosing $K^*$ later, we set

$$R_K(f)(x) = \sum_{0 \leq k \leq K} p_k + \sum_{k \in \mathbb{Z}_+ : K < k \leq K^*} G_k(p_k),$$

where $G_k(p_k)$ is given as

$$G_k(p_k) = \sum_{1 \leq i \leq n_k} \alpha_{k,j_i} M_{k,j_i}^d(x)$$

where $(\alpha_{k,j_i})_{i=1}^{|J(k)|}$ is the sorted coefficients in decreasing order of absolute value: $|\alpha_{k,j_1}| \geq |\alpha_{k,j_2}| \geq \cdots \geq |\alpha_{k,j_{|J(k)|}}|$. Then, it holds that

$$\|p_k - G_k(p_k)\|_r \leq \|p_k\|_p 2^{\delta\|k\|_{\underline{\beta}/\beta}} n_k^{-\delta},$$

where $\delta := (1/p - 1/r)$ (see the proof of Theorem 3.1 of Dũng (2011b) and Lemma 5.3 of Dũng (2011a)). Moreover, we also have

$$\|p_k\|_r \leq \|p_k\|_p 2^{\delta\|k\|_{\underline{\beta}/\beta}}$$

for $k \in \mathbb{Z}_+$ with $k > K^*$.

Here, we define $N$ as

$$N = \lceil 2^{\|K\|_{\underline{\beta}/\beta}} \rceil.$$

Let $\nu = (\widetilde{\beta} - \delta)/(2\delta)$,

$$K^* = \lceil K(1 + 1/\nu) \rceil,$$

and

$$n_k = \left\lceil 2^{\|K\|_{\underline{\beta}/\beta} - \epsilon(\|k\|_{\underline{\beta}/\beta} - \|K\|_{\underline{\beta}/\beta})} \right\rceil$$

for $k \in \mathbb{Z}_+$ with $K + 1 \leq k \leq K^*$.

Then, by Lemma 5.3 of Dũng (2011a), we have

$$\|f - R_K(f)\|_{L^r}^r \lesssim \sum_{K < k \leq K^*} \|p_k - G_k(p_k)\|_{L^r}^r + \sum_{K^* < k} \|p_k\|_{L^r}^r$$

$$\lesssim \sum_{K < k \leq K^*} [\|p_k\|_p 2^{\delta\|k\|_{\underline{\beta}/\beta}} n_k^{-\delta}]^r + \sum_{K^* < k} [2^{\delta\|k\|_{\underline{\beta}/\beta}} \|p_k\|_{L^p}]^r. \qquad (14)$$

(a) Suppose that $q \leq r$ and $r < \infty$. Then,

$$\|f - R_K(f)\|_{L^r}^q = \|f - R_K(f)\|_{L^r}^{r\frac{q}{r}}$$

$$\lesssim \left\{ \sum_{K < \|k\|_1 \leq K^*} [2^{\delta\|k\|_{\underline{\beta}/\beta}} n_k^{-\delta} \|p_k\|_{L^p}]^r + \sum_{K^* < k} [2^{\delta\|k\|_{\underline{\beta}/\beta}} \|p_k\|_{L^p}]^r \right\}^{\frac{q}{r}} \qquad (\because \text{Eq. (14)})$$

$$\lesssim \sum_{K < k \leq K^*} [2^{\delta \|k\|_{\underline{\beta}/\beta}} n_k^{-\delta} \|p_k\|_{L^p}]^q + \sum_{K^* < k} [2^{\delta \|k\|_{\underline{\beta}/\beta}} \|p_k\|_{L^p}]^q$$

$$\leq N^{-\delta q} 2^{-(\widetilde{\beta}-\delta)\|K\|_{\underline{\beta}/\beta} q} \sum_{K < k \leq K^*} [\underbrace{2^{-(\widetilde{\beta}-\delta-\delta\epsilon)(\|k\|_{\underline{\beta}/\beta} - \|K\|_{\underline{\beta}/\beta})}}_{\leq 1} 2^{\widetilde{\beta}\|k\|_{\underline{\beta}/\beta}} \|p_k\|_{L^p}]^q$$

$$+ 2^{-q(\widetilde{\beta}-\delta)\|K^*\|_{\underline{\beta}/\beta}} \sum_{K^* < k} [2^{\widetilde{\beta}\|k\|_{\underline{\beta}/\beta}} \|p_k\|_{L^p}]^q$$

$$\overset{(i)}{\lesssim} (N^{-\delta} 2^{-(\widetilde{\beta}-\delta)\|K\|_{\underline{\beta}/\beta}} + 2^{-(\widetilde{\beta}-\delta)K^*})^q \|f\|_{MB_{p,q}^s}^q \quad (\because \text{Eq. (13)})$$

$$\overset{(ii)}{\lesssim} (N^{-\widetilde{\beta}})^q \|f\|_{MB_{p,q}^\alpha}^q$$

where we used $2^{\widetilde{\beta}\|k\|_{\underline{\beta}/\beta}} \simeq 2^{\underline{\beta}k}$ in (i), and $N \simeq 2^{\|K\|_{\underline{\beta}/\beta}}$ and $\nu = (\widetilde{\beta}-\delta)/(2\delta)$ in (ii).

(b) Suppose that $q > r$ and $r < \infty$. Then, letting $\gamma = q/r \; (> 1)$ and $\gamma' = 1/(1-1/\gamma) = q/(q-r)$ (note that $\frac{1}{\gamma} + \frac{1}{\gamma'} = 1$), we have

$$\|f - R_K(f)\|_{L^r}^r \lesssim \sum_{K < k \leq K^*} [2^{\delta\|k\|_{\underline{\beta}/\beta}} n_k^{-\delta} \|p_k\|_{L^p}]^r + \sum_{K^* < k} [2^{\delta\|k\|_{\underline{\beta}/\beta}} \|p_k\|_{L^p}]^r \quad (\because \text{Eq. (14)})$$

$$\leq 2^{-\widetilde{\beta}\|K\|_{\underline{\beta}/\beta} r} \sum_{K < k \leq K^*} [2^{-(\widetilde{\beta}-\delta-\delta\nu)(\|k\|_{\underline{\beta}/\beta} - \|K\|_{\underline{\beta}/\beta})} 2^{\widetilde{\beta}\|k\|_{\underline{\beta}/\beta}} \|p_k\|_{L^p}]^r$$

$$+ \sum_{K^* < k} [2^{\widetilde{\beta}\|k\|_{\underline{\beta}/\beta}} \|p_k\|_{L^p}]^r (2^{-(\widetilde{\beta}-\delta)\|k\|_{\underline{\beta}/\beta}})^r$$

$$\leq (2^{-\widetilde{\beta}\|K\|_{\underline{\beta}/\beta}} + 2^{-(\widetilde{\beta}-\delta)\|K^*\|_{\underline{\beta}/\beta}})^r \Big\{ \sum_{K < k \leq K^*} [2^{-(\widetilde{\beta}-\delta-\delta\nu)(\|k\|_{\underline{\beta}/\beta} - \|K\|_{\underline{\beta}/\beta})} 2^{\widetilde{\beta}\|k\|_{\underline{\beta}/\beta}} \|p_k\|_{L^p}]^r$$

$$+ \sum_{K^* < k} [2^{\widetilde{\beta}\|k\|_{\underline{\beta}/\beta}} \|p_k\|_{L^p}]^r 2^{-(\widetilde{\beta}-\delta)(\|k\|_{\underline{\beta}/\beta} - \|K^*\|_{\underline{\beta}/\beta})r} \Big\}$$

$$\leq (2^{-\widetilde{\beta}\|K\|_{\underline{\beta}/\beta} r} + 2^{-(\widetilde{\beta}-\delta)\|K^*\|_{\underline{\beta}/\beta}})^r \left\{ \sum_{K < k \leq K^*} [2^{\widetilde{\beta}\|k\|_{\underline{\beta}/\beta}} \|p_k\|_{L^p}]^{r\gamma} + \sum_{K^* < k} [2^{\widetilde{\beta}\|k\|_{\underline{\beta}/\beta}} \|p_k\|_{L^p}]^{r\gamma} \right\}^{1/\gamma}$$

$$\times \left\{ \sum_{K < k \leq K^*} [2^{-(\widetilde{\beta}-\delta-\delta\nu)(\|k\|_{\underline{\beta}/\beta} - \|K\|_{\underline{\beta}/\beta})}]^{r\gamma'} + \sum_{K^* < k} [2^{-(s-\delta)(\|k\|_{\underline{\beta}/\beta} - K^*)}]^{r\gamma'} \right\}^{1/\gamma'}$$

$$\lesssim (2^{-\widetilde{\beta}\|K\|_{\underline{\beta}/\beta}} + 2^{-(\widetilde{\beta}-\delta)\|K^*\|_{\underline{\beta}/\beta}})^r \|f\|_{B_{p,q}^\beta}^r \quad (\because \text{Eq. (13) and } 2^{\widetilde{\beta}\|k\|_{\underline{\beta}/\beta}} \simeq 2^{\underline{\beta}k})$$

$$\lesssim (N^{-\widetilde{\beta}})^r \|f\|_{B_{p,q}^\beta}^r.$$

(c) Suppose that $r = \infty$. Then, similarly to the analysis in (b), we can evaluate

$$\|f - R_K(f)\|_{L^r}$$

$$\lesssim 2^{-\widetilde{\beta}\|K\|_{\underline{\beta}/\beta}} \sum_{K < k \leq K^*} [2^{-(\widetilde{\beta}-\delta-\delta\epsilon)(\|k\|_{\underline{\beta}/\beta} - \|K\|_{\underline{\beta}/\beta})} 2^{\widetilde{\beta}\|k\|_{\underline{\beta}/\beta}} \|p_k\|_{L^p}]$$

$$+ \sum_{K^* < k} [2^{\widetilde{\beta}\|k\|_{\underline{\beta}/\beta}} \|p_k\|_{L^p}] (2^{-(\widetilde{\beta}-\delta)\|k\|_{\underline{\beta}/\beta}})$$

$$\lesssim (2^{-\widetilde{\beta}\|K\|_{\underline{\beta}/\beta}} + 2^{-(\widetilde{\beta}-\delta)\|K^*\|_{\underline{\beta}/\beta}}) \|f\|_{B_{p,q}^\beta}$$

$$\lesssim N^{-\widetilde{\beta}} \|f\|_{B_{p,q}^\beta}.$$

This concludes the proof. $\qquad\qquad\qquad\qquad\qquad\qquad\qquad\qquad\qquad\qquad\qquad\qquad\square$

*Proof of Proposition 2.* We adopt the proof line employed by Suzuki (2019). Basically, we combine Lemma 1 and Lemma 2. We substitute the approximated cardinal B-spline basis $\check{M}$ into the

decomposition of $f_N$ (11). Let the set of indexes $(k,j) \in \mathbb{Z} \times \mathbb{Z}$ that consists $f_N$ given in Eq. (11) be $E_N$, i.e., $f_N = \sum_{(k,j) \in E_N} \alpha_{k,j} M_{k,j}^d$. Accordingly, we set $\check{f} := \sum_{(k,j) \in E_N} \alpha_{k,j} \check{M}_{k,j}^d$. Note that for each $x$, the number of $(k,j) \in E_N$ that satisfy $M_{k,j}(x) \neq 0$ is bounded by $(m+1)^d(1+K^*)$, and $\max_{(k,j) \in E_N} |\alpha_{k,j}| \lesssim 2^{K^* \frac{\beta}{\widetilde{\beta}}(\widetilde{\beta}-1/p)_+}$ by the norm equivalence Eq. (10). For each $x \in \mathbb{R}^d$, it holds that

$$
\begin{aligned}
|f_N(x) - \check{f}(x)| &\leq \sum_{(k,j) \in E_N} |\alpha_{k,j}||M_{k,j}^d(x) - \check{M}_{k,j}^d(x)| \\
&\leq \epsilon \sum_{(k,j) \in E_N} |\alpha_{k,j}|\mathbf{1}\{M_{k,j}^d(x) \neq 0\} \\
&\lesssim \epsilon(m+1)^d(1+K^*)2^{K^*(\underline{\beta}/\widetilde{\beta})(\widetilde{\beta}-1/p)_+}\|f\|_{B_{p,q}^s} \\
&\lesssim \epsilon \log(N) N^{(1+\nu^{-1})(\widetilde{\beta}-1/p)_+}\|f\|_{B_{p,q}^s},
\end{aligned}
$$

where we used the definition of $K^*$ in the last inequality. This evaluation yields that, for each $f \in U(B_{p,q}^\beta(\Omega))$, it holds that

$$
\|f - \check{f}\|_{L^r} \lesssim \|f - f_N\|_{L^r} + \|f_N - \check{f}\|_{L^r} \lesssim \log(N)N^{(1+\nu^{-1})(1/p-\widetilde{\beta})_+}\|f\|_{B_{p,q}^s}\epsilon + N^{-\widetilde{\beta}}.
$$

By taking $\epsilon$ to satisfy $\log(N)N^{(1+\nu^{-1})(1/p-\widetilde{\beta})_+}\epsilon \leq N^{-\widetilde{\beta}}$, we obtain the approximation error bound.

As we have seen above $\max_{(k,j) \in E_N} |\alpha_{k,j}| \lesssim 2^{K^* \frac{\beta}{\widetilde{\beta}}(\widetilde{\beta}-1/p)_+} \leq N^{(1+\nu^{-1})(1/p-\widetilde{\beta})_+}$. The max of the absolute values of parameters used in $\check{M}_{k,j}^d$ can be bounded by $2^{K^*}$ (see Suzuki (2019)) which is bounded by $N^{d(1+\nu^{-1})(1/p-\widetilde{\beta})_+}$. Then, we obtain the assertion.

$\square$

## B.1 Proof of Theorem 6 and Theorem 1

*Proof of Theorem 6.* This proof is almost obvious from Proposition 2. We know that, from Proposition 2, for $g \in U(B_{p,q}^\beta([0,1]^{\tilde{d}}))$, there exists $\check{f} \in \Phi(L_1(\tilde{d}), W_1(\tilde{d}), S_1(\tilde{d}), B_1(\tilde{d}))$ such that

$$
\|\check{f} - g\|_r \lesssim N^{-\widetilde{\beta}}.
$$

Because the density of the distribution of $Ax + b$ is bounded above when $x$ obeys the uniform distribution on $\Omega$, this also yields

$$
\|\check{f} \circ (A \cdot +b) - g \circ (A \cdot +b)\|_r \lesssim N^{-\widetilde{\beta}}.
$$

(note that the Lebesgue measure on $\Omega = [0,1]^d$ corresponds to the uniform distribution on $\Omega$). If $\check{f}$ can be written as

$$
\check{f}(x) = (\mathcal{W}^{(L_1)}\eta(\cdot) + b^{(L)}) \circ \cdots \circ (\mathcal{W}^{(1)}x + b^{(1)}),
$$

then we have

$$
\check{f} \circ (A \cdot +b) = (\mathcal{W}^{(L_1)}\eta(\cdot) + b^{(L)}) \circ \cdots \circ (\mathcal{W}^{(1)}A \cdot +b^{(1)} + \mathcal{W}^{(1)}b) \in \Phi(L_1(\tilde{d}), W_1(\tilde{d}), S_1(\tilde{d}), (\tilde{d}C+1)B_1(\tilde{d})).
$$

$\square$

*Proof of Theorem 1.*

$$
\mathcal{H}_{\text{deep}} := \{h_H \circ \cdots \circ h_1(x) \mid h_\ell : [0,1]^{m_\ell} \rightarrow [0,1]^{m_{\ell+1}}, \ h_{\ell,k} \in U(B_{p,q}^{\beta^{(\ell)}}([0,1]^{m_\ell})) \ (\forall k \in [m_{\ell+1}])\}.
$$

Since $\widetilde{\beta}^{(\ell)} > 1/p$, we can show that for each $h_{\ell,k}$, there exists $\check{f}_{\ell,k} \in \Phi(L_1(m_\ell), W_1(m_\ell), S_1(m_\ell), B_1(m_\ell))$ such that

$$
\|\check{f}_{\ell,k} - h_{\ell,k}\|_\infty \lesssim N^{-\widetilde{\beta}}.
$$

Moreover, from the proof of Proposition 2, we can share all parameters other than the last layer among $\check{f}_{\ell,k}$ $(k = 1, \ldots, m_{\ell+1})$. If necessary, we may modify $\check{f}_{\ell,k}$ so that $0 \leq \check{f}_{\ell,k}(x) \leq 1$ $(\forall x \in$

$[0,1]^{m_\ell}$) by adding one additional clipping layer which can be realized by ReLU (actually, the clipping operator can be constructed by a linear combination of 2 nodes with ReLU activation as $f(x) = \max\{x, 0\} - \max\{x - 1, 0\} = \min\{\max\{x, 0\}, 1\}$ for $x \in \mathbb{R}$). The approximation error of the whole layer can be evaluated as

$$\|h_H \circ \cdots \circ h_1 - \check{f}_H \circ \cdots \circ \check{f}_1\|_\infty$$

$$\leq \sum_{\ell=1}^{H} \|h_H \circ \cdots \circ h_{\ell+1} \circ h_\ell \circ \check{f}_{\ell-1} \circ \cdots \circ \check{f}_1 - h_H \circ \cdots \circ h_{\ell+1} \circ \check{f}_\ell \circ \check{f}_{\ell-1} \circ \cdots \circ \check{f}_1\|_\infty$$

$$\leq \sum_{\ell=1}^{H} \|h_H \circ \cdots \circ h_{\ell+1} \circ h_\ell - h_H \circ \cdots \circ h_{\ell+1} \circ \check{f}_\ell\|_\infty.$$

Proposition 1 tells that $h_{\ell',k} \in \mathcal{C}^{(\underline{\beta}^{(\ell')} - 1/p) \wedge 1}$; thus, $h_{\ell',k}$ is $\gamma_{\ell'}$-Hölder continuous where $\gamma_{\ell'} := (\underline{\beta}^{(\ell')} - 1/p) \wedge 1$. Their composition $h_H \circ h_{H-1} \circ \cdots \circ h_{\ell+1}$ is $\Gamma_\ell$-Hölder continuous where $B_\ell = \prod_{\ell'=\ell+1}^{H} \gamma_{\ell'}$. Therefore, we have

$$\|h_H \circ \cdots \circ h_{\ell+1} \circ h_\ell - h_H \circ \cdots \circ h_{\ell+1} \circ \check{f}_\ell\|_\infty \lesssim \|h_\ell - \check{f}_\ell\|_\infty^{B_\ell},$$

where $\|\cdot\|_\infty$ for a vector-valued function $g : \mathbb{R}^{d'} \to \mathbb{R}^{d''}$ is defined as $\sup_x \|g(x)\|$. Summing up this evaluation for $\ell = 1, \ldots, H$ concludes that

$$\|h_H \circ \cdots \circ h_1 - \check{f}_H \circ \cdots \circ \check{f}_1\|_\infty \lesssim \sum_{\ell=1}^{H} N^{-B_\ell \widetilde{\beta}^{(\ell)}} \lesssim \max_{\ell \in H} N^{-\widetilde{\beta}^{*(\ell)}}.$$

Consequently, the whole network can be realized as an element of $\Phi(L, W, S, B)$ where

$$L = \sum_{\ell=1}^{H} (L_1(m_\ell) + 1), \ W = \max_\ell (W_1(m_\ell) \vee m_{\ell+1}),$$

$$S = \sum_{\ell=1}^{H} (S_1(m_\ell) + 3m_{\ell+1}), \ B = \max_\ell B_1(m_\ell).$$

$\square$

## C  Proofs of estimation error bound (Theorem 2 and Theorem 3 )

*Proof of Theorem 2.* We follow the proof strategy from Schmidt-Hieber (2018); Suzuki (2019) which uses Proposition 4. It suffices to the covering number of $\hat{\mathcal{F}} = \{\bar{f} \mid f \in \Psi(L, W, S, B)\}$ for $(L, W, S, B)$ given in Theorem 6 where $\bar{f}$ is the clipped version of a function $f$. Note that the covering number of $\hat{\mathcal{F}}$ is not larger than that of $\Psi(L, W, S, B)$. Hence, it is sufficient to evaluate that of $\Psi(L, W, S, B)$. From Lemma 6, the covering number of this class is upper bounded by

$$\log N(\delta, \hat{\mathcal{F}}, \|\cdot\|_\infty) \lesssim N \log(N)[\log(N)^2 + \log(\delta^{-1})].$$

From Proposition 2, there exists $\check{f} \in \Phi(L, W, S, B)$ such that

$$\|f^\circ - R_K(f^\circ)\|_2 \lesssim N^{-\widetilde{\beta}}.$$

Moreover, we notice that $\|f - f^\circ\|_{L^2(P_X)}^2 \leq R\|f - f^\circ\|_2^2$. for any $f : [0,1]^d \to \mathbb{R}$ because the density $p_X$ of $P_X$ is bounded by $R$. Therefore, by applying Proposition 4 with $\delta = 1/n$, we have

$$\mathrm{E}_{D_n}[\|\hat{f} - f^\circ\|_{L^2(P_X)}^2] \lesssim N^{-2\widetilde{\beta}} + \frac{N \log(N)(\log(N)^2 + \log(n))}{n} + \frac{1}{n}.$$

Here, we can minimize the right hand side by setting $N \asymp n^{\frac{1}{2\widetilde{\beta}+1}}$ up to $\log(n)^3$-order, and then we obtain the estimation error of the least squares estimator as

$$n^{-\frac{2\widetilde{\beta}}{2\widetilde{\beta}+1}} \log(n)^3.$$

This yields the assertion. $\square$

*Proof of Theorem 3.* The proof is almost identical to the proof of Theorem 2, except that we use Theorem 1 as an approximation error bound. $\square$

## D   Embedding theorem

**Lemma 3.** *For $0 < p^{(1)}, p^{(2)} \leq \infty$, let $\beta^{(1)}, \beta^{(2)} \in \mathbb{R}_{++}^d$ such that they satisfy*

$$\tilde{\beta}^{(1)} - \tilde{\beta}^{(2)} \geq \frac{1}{p^{(1)}} - \frac{1}{p^{(2)}}, \tag{15}$$

$$\beta^{(2)} = \gamma \beta^{(1)},$$

$$p^{(1)} < p^{(2)},$$

*for $0 < \gamma < 1$. Then, it holds that*

$$B_{p^{(1)},q}^{\beta^{(1)}} \hookrightarrow B_{p^{(2)},q}^{\beta^{(2)}}.$$

*Proof.* We show the assertion only for the situation where $p^{(1)} \neq \infty$, $p^{(2)} \neq \infty$, and $q \neq \infty$. The proof for the setting in which $p^{(1)} = \infty$, $p^{(2)} = \infty$, or $q = \infty$ is satisfied is almost identical. Recall the following norm equivalence shown in Lemma 2:

$$\|f\|_{B_{p,q}^s} \simeq \|(\alpha_{k,j})_{k,j}\|_{b_{p,q}^\beta} = \left\{ \sum_{k=0}^\infty \left[ 2^{k[\underline{\beta} - (\sum_{i=1}^d \lfloor k\beta_i' \rfloor /k)/p]} \left( \sum_{j \in J(k)} |\alpha_{k,j}|^p \right)^{1/p} \right]^q \right\}^{1/q},$$

when $p, q < \infty$. Since $\frac{p^{(1)}}{p^{(2)}} < 1$, it holds that

$$\left( \sum_{j \in J(k)} |\alpha_{k,j}|^{p^{(1)}} \right)^{1/p^{(1)}} = \left( \sum_{j \in J(k)} |\alpha_{k,j}|^{p^{(2)} \frac{p^{(1)}}{p^{(2)}}} \right)^{1/p^{(1)}}$$

$$\geq \left( \sum_{j \in J(k)} |\alpha_{k,j}|^{p^{(2)}} \right)^{\frac{p^{(1)}}{p^{(2)}} \frac{1}{p^{(1)}}} = \left( \sum_{j \in J(k)} |\alpha_{k,j}|^{p^{(2)}} \right)^{\frac{1}{p^{(2)}}}.$$

Moreover, we have

$$2^{k[\underline{\beta}^{(1)} - (\sum_{i=1}^d \lfloor k\beta_i'^{(1)} \rfloor /k)/p^{(1)}]}$$

$$\simeq 2^{k\underline{\beta}^{(1)} - \sum_{i=1}^d \beta_i'^{(1)} /p^{(1)}} = 2^{k \frac{\beta^{(1)}}{\tilde{\beta}^{(1)}} \left( \tilde{\beta}^{(1)} - \frac{1}{p^{(1)}} \right)} \overset{(a)}{=} 2^{k \frac{\beta^{(2)}}{\tilde{\beta}^{(2)}} \left( \tilde{\beta}^{(1)} - \frac{1}{p^{(1)}} + \frac{1}{p^{(2)}} - \frac{1}{p^{(2)}} \right)}$$

$$\overset{(b)}{\geq} 2^{k \frac{\beta^{(2)}}{\tilde{\beta}^{(2)}} \left( \tilde{\beta}^{(2)} - \frac{1}{p^{(2)}} \right)} = 2^{k\underline{\beta}^{(2)} - \sum_{i=1}^d \beta_i'^{(2)} /p^{(2)}} \simeq 2^{k[\underline{\beta}^{(2)} - (\sum_{i=1}^d \lfloor k\beta_i'^{(2)} \rfloor /k)/p^{(2)}]},$$

where we used the condition $\beta^{(2)} = \gamma \beta^{(1)}$ in (a), and we used the condition from Eq. (15) in (b). These relations yield the following evaluation:

$$\|f\|_{B_{p^{(1)},q}^{\beta^{(1)}}}$$

$$\simeq \|(\alpha_{k,j})_{k,j}\|_{b_{p^{(1)},q}^{\beta^{(1)}}}$$

$$= \left\{ \sum_{k=0}^\infty \left[ 2^{k[\underline{\beta}^{(1)} - (\sum_{i=1}^d \lfloor k\beta_i'^{(1)} \rfloor /k)/p^{(1)}]} \left( \sum_{j \in J(k)} |\alpha_{k,j}|^{p^{(1)}} \right)^{1/p^{(1)}} \right]^q \right\}^{1/q}$$

$$\gtrsim \left\{ \sum_{k=0}^\infty \left[ 2^{k[\underline{\beta}^{(2)} - (\sum_{i=1}^d \lfloor k\beta_i'^{(2)} \rfloor /k)/p^{(2)}]} \left( \sum_{j \in J(k)} |\alpha_{k,j}|^{p^{(2)}} \right)^{1/p^{(2)}} \right]^q \right\}^{1/q}$$

$$\simeq \|f\|_{B_{p^{(2)},q}^{\beta^{(2)}}}.$$

This yields the assertion. □

By combining Lemma 3 with the relation $B_{\infty,\infty}^{\gamma\beta} \hookrightarrow \mathcal{C}^{\gamma\underline{\beta}}$ (Triebel, 2011), we immediately obtain the following corollary.

**Corollary 1.** *Suppose that $\widetilde{\beta} > p$, then for $\gamma = \frac{\widetilde{\beta} - p}{\widetilde{\beta}}$, it holds that*

$$B_{p,q}^\beta \hookrightarrow B_{\infty,q}^{\gamma\beta} \hookrightarrow B_{\infty,\infty}^{\gamma\beta} \hookrightarrow \mathcal{C}^{\gamma\underline{\beta}}.$$

# E Minimax optimality

In this section, we demonstrate the proof of Theorem 4. Before this, we prepare the basic notions. The $\epsilon$-covering number $\mathcal{N}(\epsilon, \mathcal{C}, \hat{d})$ of a metric space $\mathcal{C}$ equipped with a metric $\hat{d}$ that is the minimal number of balls with radius $\epsilon$ measured by the metric $\hat{d}$ required to cover the set $\mathcal{C}$ (van der Vaart & Wellner, 1996). Similarly, the $\delta$-packing number $\mathcal{M}(\delta, \mathcal{C}, \hat{d})$ is defined as the largest number of elements $\{f_1, \ldots, f_{\mathcal{M}}\} \subseteq \mathcal{C}$ such that $\hat{d}(f_i, f_j) \geq \delta$ for all $i \neq j$.

Raskutti et al. (2012) showed the following inequality in their proof of Theorem 2(b) by utilizing the result by Yang & Barron (1999).

**Lemma 4.** *Let $\mathcal{F}^\circ$ be the model of the true function. For a given $\delta_n > 0$ and $\varepsilon_n > 0$, let $Q$ be the $\delta_n$-packing number $\mathcal{M}(\delta_n, \mathcal{F}^\circ, L^2(P_X))$ of $\mathcal{F}^\circ$ and $N$ be the $\varepsilon_n$ covering number of that. Suppose that they satisfy the following condition:*

$$\frac{n}{2\sigma^2}\varepsilon_n^2 \leq \log(N),$$
$$8\log(N) \leq \log(Q), \ 4\log(2) \leq \log(Q). \tag{16}$$

*Then, the minimax learning rate is lower bounded as*

$$\inf_{\widehat{f}} \sup_{f^* \in \mathcal{F}^\circ} \mathrm{E}_{D_n}[\|\widehat{f} - f^*\|_{L^2(P_X)}^2] \geq \frac{\delta_n^2}{4}.$$

*This concludes the assertion.*

Now, we are ready to show Theorem 4.

*Proof of Theorem 4.* Proposition 10 of Triebel (2011) showed that the $\epsilon$-covering number of the unit ball of anisotropic Besov spaces $B_{p,q}^\beta(\Omega)$ can be evaluated as

$$\log \mathcal{N}(\epsilon, U(B_{p,q}^\beta(\Omega)), \|\cdot\|_r) \simeq \epsilon^{-1/\widetilde{\beta}},$$

for $0 < p, q \leq \infty$, $1 \leq r < \infty$, and $\beta \in \mathbb{R}_{++}^d$ that satisfy

$$\widetilde{\beta} > \max\left\{\frac{1}{p} - \frac{1}{r}, \frac{1}{p} - 1, 0\right\}.$$

**Affine composition model**:

Apparently, $U(B_{p,q}^\beta(\Omega))$ is included in $\mathcal{H}_{\mathrm{aff}}$. Hence, noting that $P_{\mathcal{X}}$ is the uniform distribution and $\|\cdot\|_2 = \|\cdot\|_{L^2(P_X)}$, the covering number of $\mathcal{H}_{\mathrm{aff}}$ can be lower bounded by

$$\log \mathcal{N}(\mathcal{H}_{\mathrm{aff}}, \|\cdot\|_{L^2(P_X)}) \gtrsim \epsilon^{-1/\widetilde{\beta}}.$$

From this evaluation, Lemma 4 yields that there exists $C_1 > 0$ independent of $n$ such that

$$\inf_{\widehat{f}} \sup_{f^* \in \mathcal{H}_{\mathrm{aff}}} \mathrm{E}_{D_n}[\|\widehat{f} - f^*\|_{L^2(P_X)}^2] \geq C_1 n^{-\frac{2\widetilde{\beta}}{2\widetilde{\beta}+1}}.$$

To see this, we may just set $\epsilon_n \simeq \delta_n \simeq n^{-\frac{2\widetilde{\beta}}{2\widetilde{\beta}+1}}$ in Eq. (16) of Lemma 4.

**Deep composition model**:

Next, we show the minimax rate for the deep composition model. Basically, we follow the same strategy developed by Schmidt-Hieber (2018), but we need to modify some technical details because we are dealing with anisotropic Besov spaces while Schmidt-Hieber (2018) analyzed isotropic Hölder space. Let $\ell^* := \min_{\ell \in [H]} \widetilde{\beta}^{*(\ell)}$, and $s^{(\ell)} := (\beta^{(\ell)} - 1/p + \epsilon) \wedge 1$ where $\epsilon > 0$ can be arbitrary small for $q < \infty$ and $\epsilon = 0$ for $q = \infty$. Without loss of generality, we may assume that $\beta_1^{(\ell)} \leq \beta_2^{(\ell)} \leq \cdots \leq \beta_d^{(\ell)}$ for $\ell \in [H]$. Let us consider a sub-model $\mathcal{H}_{\mathrm{deep}}'$ of $\mathcal{H}_{\mathrm{deep}}$ defined as

$$\mathcal{H}_{\mathrm{deep}}' := \{g_H \circ \cdots \circ g_1 \mid$$

$$g_\ell(x) = x \quad (\ell = 1, \ldots, \ell^* - 1),$$
$$g_{\ell^*}(x) = (g_{\ell^*,1}(x), 0, \ldots, 0)^\top \text{ where } g_{\ell^*,1} \in U(B_{p,q}^\beta(\Omega)),$$
$$g_\ell(x) = (x_1^{s^{(\ell)}}, 0, \ldots, 0)^\top \quad (\ell = \ell^* + 1, \ldots, H)\}.$$

For $\ell = \ell^* + 1, \ldots, H$, through a cumbersome calculation, we can verify that $x_1^{s^{(\ell)}} \in B_{p,q}^{\beta^{(\ell)}}([0,1])$ for $x \in [0,1]$, which ensures $g_{\ell,j}(x) \in B_{p,q}^{\beta^{(\ell)}}([0,1]^d)$ for $j = 1, \ldots, d$. To lower bound the covering number, we concretely construct a subset the cardinality of which can be easily estimated. For that purpose, we use the expansion $f = \sum_{k=0}^\infty \sum_{j \in J(k)} \alpha_{k,j} M_{k,j}^d(x)$ and the norm equivalence $\|f\|_{B_{p,q}^\beta} \simeq \|(\alpha_{k,j})_{k,j}\|_{b_{p,q}^\beta}$ given in Lemma 2. For a while, we let $\beta := \beta^{(\ell^*)}$ and $B := \prod_{q=\ell^*+1}^H s^{(\ell)}$. We define $k \in \mathbb{N}$ so that $k$ satisfies $2^{k\frac{\beta}{\widetilde{\beta}}} \simeq n^{\frac{1}{1+2B\widetilde{\beta}}}$. For this choice of $k$, take a subset $\hat{J}(k) \subset J(k)$ such that $|\hat{J}(k)| \simeq |J(k)|$ and for each $j, j' \in \hat{J}(k)$ with $j \neq j'$, the supports of $M_{k,j}^d$ and $M_{k,j'}^d$ are disjoint. Using this index set $\hat{J}(k)$, we consider a set of functions that is given by

$$\hat{\mathcal{H}}_{\ell^*} := \left\{ f = \sum_{j \in \hat{J}(k)} \alpha_{k,j} M_{k,j}^d(x) \mid \alpha_{k,j} \in \{0, 2^{-k\underline{\beta}}\} \right\}.$$

We can check that $|\hat{\mathcal{H}}_{\ell^*}| = |\hat{J}(k)| \simeq 2^{k\sum_{j=1}^d \beta_j'} = 2^{k\underline{\beta}/\widetilde{\beta}}$ and $\|f\|_{B_{p,q}^\beta} \lesssim 1$ for all $f \in \hat{\mathcal{H}}_{\ell^*}$ from the norm equivalence (10). For any $g_w = \sum_{j \in \hat{J}(k)} w_j 2^{-k\underline{\beta}} M_{k,j}^d(x) \in \hat{\mathcal{H}}_{\ell^*}$ ($w \in \{0,1\}^{|\hat{J}_k|}$), we can see that

$$f_w(x) = g_H \circ \cdots \circ g_{\ell^*+1} \circ g_w \circ g_{\ell^*-1} \circ \cdots \circ g_1(x)$$
$$= \sum_{j \in \hat{J}(k)} w 2^{-Bk\underline{\beta}} M_{k,j}^{dB}(x).$$

If $w \neq w'$, then we can see that

$$\|f_w - f_{w'}\|_{L^2(P_X)}^2 \gtrsim \text{Ham}(w, w') 2^{-2BK\underline{\beta}} 2^{-k\underline{\beta}/\widetilde{\beta}}$$
$$\gtrsim \text{Ham}(w, w') 2^{-k\underline{\beta}(2B\widetilde{\beta}+1)/\widetilde{\beta}},$$

where Ham is the Hamming distance because $\|M_{k,j}^d\|_{L^2(P_X)}^2 \simeq 2^{-k\underline{\beta}/\widetilde{\beta}}$.

Then, by the Varshamov–Gilbert bound (see Lemma 2.9 of Tsybakov (2008), for example), there exists a subset $W_k \subset \{0,1\}^{|\hat{J}(k)|}$ such that $|W_k| \geq 2^{|\hat{J}(k)|/8}$ and $\text{Ham}(w, w') \geq |\hat{J}(k)|/8$ for all $w, w' \in W_k$ with $w \neq w'$. This yields

$$\|f_w - f_{w'}\|_{L^2(P_X)}^2 \gtrsim 2^{k\underline{\beta}/\widetilde{\beta}} 2^{-k\underline{\beta}(2B\widetilde{\beta}+1)/\widetilde{\beta}} = 2^{-2Bk\underline{\beta}} \simeq n^{-\frac{2B\widetilde{\beta}}{2B\widetilde{\beta}+1}},$$

where the definition of $k$ is used. This implies that there exists a subset $\mathcal{H}''_{\text{deep}} \subset \mathcal{H}'_{\text{deep}} (\subset \mathcal{H}_{\text{deep}})$ such that

$$\log(\mathcal{N}(\epsilon_n, \mathcal{H}''_{\text{deep}}, \|\cdot\|_{L^2(P_X)})) \gtrsim n^{\frac{1}{1+2B\widetilde{\beta}}}$$

for $\epsilon_n \gtrsim n^{-\frac{B\widetilde{\beta}}{2B\widetilde{\beta}+1}}$. Then, by Lemma 4, we obtain that the minimax optima rate on $\mathcal{H}_{\text{deep}}$ is lower bounded as

$$\inf_{\widehat{f}} \sup_{f^* \in \mathcal{F}^\circ} \mathbb{E}_{D_n}[\|\widehat{f} - f^*\|_{L^2(P_X)}^2] \gtrsim n^{-\frac{B\widetilde{\beta}}{2B\widetilde{\beta}+1}}.$$

$\square$

## F   Minimax optimal rate of linear estimators

Define the convex hull of a function class $\mathcal{F}^\circ$ as

$$\text{conv}(\mathcal{F}^\circ) := \left\{ f(x) = \sum_{j=1}^M \lambda_j f_j(x) \mid M = 1, 2, \ldots, \ f_j \in \mathcal{F}^\circ, \ \lambda_j \geq 0, \ \sum_{j=1}^M \lambda_j = 1 \right\}.$$

Let $\overline{\mathrm{conv}}(\cdot)$ is the closure of $\overline{\mathrm{conv}}(\cdot)$ with respect to $L_2(P_X)$-norm.

**Proposition 3** (Hayakawa & Suzuki (2019)). *The minimax optimal rate of linear estimators on a target function class $\mathcal{F}^\circ$ is the same as that on the convex hull of $\mathcal{F}^\circ$:*

$$\inf_{\widehat{f}:\ linear} \sup_{f^\circ \in \mathcal{F}^\circ} \mathrm{E}_{D_n}[\|f^\circ - \widehat{f}\|^2_{L^2(P_X)}] = \inf_{\widehat{f}:\ linear} \sup_{f^\circ \in \overline{\mathrm{conv}}(\mathcal{F}^\circ)} \mathrm{E}_{D_n}[\|f^\circ - \widehat{f}\|^2_{L^2(P_X)}].$$

See Hayakawa & Suzuki (2019) for the proof of this proposition.

*Proof of Theorem 5.* We basically follow the strategy developed by Zhang et al. (2002). Let $\mu$ be the uniform measure on $\Omega$. They essentially showed the following statement in their Theorem 1. Suppose that the space $\Omega$ has even partition $\mathcal{A}$ such that $|\mathcal{A}| = 2^K$ for an integer $K \in \mathbb{N}$, each $A$ has equivalent measure $\mu(A) = 2^{-K}$ for all $A \in \mathcal{A}$, and $\mathcal{A}$ is indeed a partition of $\Omega$, i.e., $\cup_{A \in \mathcal{A}} = \Omega$, $A \cap A' = \emptyset$ for $A, A' \in \Omega$ and $A \neq A'$. Then, if $K$ is chosen as $n^{-\gamma_1} \leq 2^{-K} \leq n^{-\gamma_2}$ for constants $\gamma_1, \gamma_2 > 0$ that are independent of $n$, then there exists an event $\mathcal{E}$ such that, for a constant $C' > 0$,

$$|\{x_i \mid x_i \in A \ (i \in \{1, \dots, n\})\}| \leq C'n/2^K \quad (\forall A \in \mathcal{A}),$$
$$P(\mathcal{E}) \geq 1 + o(1).$$

We call this property of $\mathcal{A}$ "Condition A."

Here, we consider a set $\mathcal{F}^\circ$ of functions on $\Omega$ for which there exists $\Delta > 0$ that satisfies the following conditions:

1. There exists $F > 0$ such that, for any $A \in \mathcal{A}$, there exists $g \in \mathcal{F}^\circ$ that satisfies $g(x) \geq \frac{1}{2}\Delta F$ for all $x \in A$,

2. There exists $K'$ and $C'' > 0$ such that $\frac{1}{n}\sum_{i=1}^n g(x_i)^2 \leq C''\Delta^2 2^{-K'}$ for any $g \in \mathcal{F}^\circ$ on the event $\mathcal{E}$.

We call this condition of the function class $\mathcal{F}^\circ$ "Condition B."

Let the minimax optimal rate of linear estimators on the function class $\mathcal{F}^\circ$ be

$$R^* = \inf_{\widehat{f}:\mathrm{linear}} \sup_{f^\circ \in \mathcal{F}^\circ} \mathrm{E}_{D^n}[\|\widehat{f} - f^\circ\|^2_{L^2(P_X)}].$$

Then, under Conditions A and B, there exists a constant $F_1$ such that at least one of the following inequalities holds:

$$\frac{F^2}{4F_1 C''}\frac{2^{K'}}{n} \leq R^*, \tag{17a}$$

$$\frac{F^3}{32}\Delta^2 2^{-K} \leq R^*, \tag{17b}$$

for sufficiently large $n$.

*(i) Proof of Eq. (6).*

For given $k \in \mathbb{N}$ (which will be fixed later), let $\Delta = 2^{-k[\underline{\beta} - (\sum_{i=1}^d \lfloor k\beta_i'\rfloor/k)/p]}$. Then, from the wavelet expansion of anisotropic Besov space (9),

$$f_w = \sum_{j \in J(k)} \Delta w_j M_{k,j}^d(x) \in CU(B_{p,q}^\beta(\Omega)),$$

where $C > 0$ is a constant and $w = (w_j)_{j \in J(k)}$ is a one-hot vector, i.e., $w_j = 1$ for some $j \in J(k)$ and $w_{j'} = 0$ for all $j' \in J(k)$ with $j' \neq j$. This expansion ensures that, for $K = \sum_{i=1}^d \lfloor k\beta_i'\rfloor$, there exists a partition $\mathcal{A}$ of $\Omega$ that satisfies Condition A, and for any $A \in \mathcal{A}$, there exists $w$ such that $f_w(x) \gtrsim \Delta$ for all $x \in A$ and

$$\frac{1}{n}\sum_{i=1}^n f_w(x_i)^2 \leq \frac{1}{n}\Delta^2|\{i \mid x_i \in A \ (i = 1, \dots, n)\}| \lesssim \Delta^2 2^{-K},$$

on the event $\mathcal{E}$, which ensures that $\mathcal{F}^\circ = \{f_w \mid w \text{ is a one-hot vector}\}$ satisfies Condition B. Hence, by choosing $k \in \mathbb{N}$ so that $2^K \simeq n^{\frac{1}{2(\tilde\beta + \frac{1}{2} - \frac{1}{p}) + 1}}$ (recall that $K = \sum_{i=1}^d \lfloor k\beta_i' \rfloor$ by definition), and setting $K = K'$, then Eq. (17) gives

$$R^* \gtrsim n^{-\frac{2\tilde\beta - v}{2\tilde\beta - v + 1}},$$

for $v = 2(1/p - 1/2)$. This yields the assertion because $\mathcal{F}^\circ \subset CU(B_{p,q}^\beta(\Omega))$ for a constant $C$.

*(ii) Proof of Eq. (7).*

Let $\beta^* := \overline\beta = \beta_1 = \cdots = \beta_{\tilde d} = \underline\beta$. For $m$ such that $\beta^* < \min\{m, m - 1 + 1/p\}$, let $\phi_{\tilde d}(x) = \prod_{j=1}^{\tilde d} \mathcal{N}_m(x_i - (m+1)/2)$ $(x \in \mathbb{R}^{\tilde d})$.

**(ii-a)** *Setting of $\tilde d \geq d/2$:*

Let $V_{\tilde d, d} := \{U \in \mathbb{R}^{\tilde d \times d} \mid UU^\top = I_{\tilde d}\}$ be the Stiefel manifold and let $\pi_{V_{\tilde d, d}}$ be the invariant measure on the Stiefel manifold (i.e., the uniform distribution). Then, let $\bar\phi_{\tilde d} : \mathbb{R}^d \to \mathbb{R}$ be

$$\bar\phi_{\tilde d}(x) = \int \phi_{\tilde d}(Ux) \mathrm{d}\pi_{V_{\tilde d, d}}(U) \quad (x \in \mathbb{R}^d).$$

We can see that $\bar\phi_{\tilde d}$ is spherically symmetric and there exists $F, C > 0$ such that

$$\bar\phi_{\tilde d}(x) \geq F \quad (\forall x \in \mathbb{R}^d \text{ s.t. } \|x\| \leq 1),$$

and

$$\bar\phi_{\tilde d}(x) \leq \begin{cases} C\|x\|^{-\tilde d} & (\|x\| \geq 1), \\ 1 & (\|x\| \leq 1). \end{cases}$$

The last inequality can be checked by the fact that for a sufficiently large $R > 0$, the measure of the set $\mu_R(\{x \mid \|x\| = R, \phi_{\tilde d}(x) > 0\}) \lesssim 1 \times R^{d - \tilde d - 1}/R^{d-1} = R^{-\tilde d}$ (here, $\mu_R$ is the uniform probability measure on the sphere $S_{d-1}(R) = \{x \in \mathbb{R}^d \mid \|x\| = R\}$) and $\|\phi_{\tilde d}\|_\infty \leq 1$.

By the construction of $\phi_{\tilde d}$ and the wavelet expansion of anisotropic Besov space (9) with the norm equivalence (10), we have that there exists a constant $c > 0$ such that, for any $k \in \mathbb{N}$ and $\bar b = \left[\frac{1}{2} - 2^{-k}\left(\frac{m+1}{2} - \lfloor \frac{m+1}{2} \rfloor\right)\right](1, \ldots, 1)^\top \in \mathbb{R}^{\tilde d}$, it holds that

$$c\Delta\phi_{\tilde d}\left(2^k(\cdot - \bar b)\right) \in U(B_{p,q}^{\beta^*}([0,1]^{\tilde d})),$$

where $\Delta = 2^{-k(\beta^* - \tilde d/p)}$. Here, let $0 < \bar c < 1$ be a constant such that $\bar c U(x - b') + \bar b \in [0,1]^{\tilde d}$ for any $x, b' \in [0,1]^d$ and any $U \in V_{\tilde d, d}$. Then, we have that, for any $b' \in [0,1]^d$,

$$c\Delta\phi_{\tilde d}(2^k \bar c U(\cdot - b')) = c\Delta\phi_{\tilde d}(2^k(\cdot - \bar b)) \circ (\bar c U(\cdot - b') + \bar b) \in \mathcal{H}_{\mathrm{aff}},$$

for any $U \in V_{\tilde d, d}$. By the convex hull argument (Proposition 3), this yields that

$$R_*^{\mathrm{lin}}(\mathcal{H}_{\mathrm{aff}}) = R_*^{\mathrm{lin}}(\overline{\mathrm{conv}}(\mathcal{H}_{\mathrm{aff}})) \geq R_*^{\mathrm{lin}}(\{c\Delta\bar\phi_{\tilde d}(2^k \bar c(\cdot - b')) \mid b' \in \Omega\}).$$

Hence, it suffices to lower bound the far right-hand side of this inequality. We consider a partition $\mathcal{A}$ of $\Omega$, where $A \in \mathcal{A}$ has the form $A = [2^{-k}j_1, 2^{-k}(j_1 + 1)] \times \cdots \times [2^{-k}j_d, 2^{-k}(j_d + 1)]$ for $0 \leq j_i \leq 2^k - 1$ $(i = 1, \ldots, d)$. Let $\hat J(k) = \{(j_1, \ldots, j_d) \mid 0 \leq j_i \leq 2^{k-1}\}$ and $A_j = [2^{-k}j_1, 2^{-k}(j_1 + 1)] \times \cdots \times [2^{-k}j_d, 2^{-k}(j_d + 1)] \in \mathcal{A}$ for $j \in \hat J(k)$. Let $\varphi_{A_j} = c\bar\phi_{\tilde d}(2^k \bar c(\cdot - b_{A_j}))$, where $b_{A_j} = (2^{-k}(j_1 + 1/2), \ldots, 2^{-k}(j_d + 1/2))^\top$ for $j \in \hat J(k)$. We can see that $|\mathcal{A}| = 2^{dk}$. Hence, $\mathcal{A}$ satisfies Condition A with $K = dk$ if $2^k$ is in polynomial order with respect to $n$.

Moreover, there exists $F > 0$ such that $\varphi_A(x) \geq F$ for all $x \in A$. Next, we evaluate $\frac{1}{n}\sum_{i=1}^n \varphi_A(x_i)^2$. On the event $\mathcal{E}$, there exists $C'$ such that $|\{i \in [n] \mid x_i \in A'\}| \leq C'n/2^K = C'n\mu(A')$ for all $A' \in \mathcal{A}$. Here, let

$$\bar\varphi_A(x) := \begin{cases} cC\|2^k \bar c(x - b_A)\|^{-\tilde d} & (\|2^k \bar c(x - b_A)\| \geq 1), \\ c & (\text{otherwise}), \end{cases}$$

then $\bar{\varphi}_A(x) \geq \varphi_A(x)$. Thus, we can upper bound $\frac{1}{n}\sum_{i=1}^n \varphi_A(x_i)^2$ as

$$\frac{1}{n}\sum_{i=1}^n \varphi_A(x_i)^2 \leq \frac{1}{n}\sum_{i=1}^n \bar{\varphi}_A(x_i)^2 = \frac{1}{n}\sum_{A'\in\mathcal{A}}\sum_{x_i\in A'}\bar{\varphi}_A(x_i)^2 \leq \frac{1}{n}\sum_{A'\in\mathcal{A}}C'\frac{n}{2^K}\max_{x\in A'}\bar{\varphi}_A(x)^2$$

$$= C'\sum_{A'\in\mathcal{A}}\mu(A')\max_{x\in A'}\bar{\varphi}_A(x)^2 = C'\sum_{A'\in\mathcal{A}}\mu(A')\min_{x\in A'}\bar{\varphi}_A(x)^2\frac{\max_{x\in A'}\bar{\varphi}_A(x)^2}{\min_{x\in A'}\bar{\varphi}_A(x)^2}$$

$$\leq C'\sum_{A'\in\mathcal{A}}\mu(A')\min_{x\in A'}\bar{\varphi}_A(x)^2\frac{\max_{x\in A'}\bar{\varphi}_A(x)^2}{\min_{x\in A'}\bar{\varphi}_A(x)^2}$$

$$\leq C'\sum_{A'\in\mathcal{A}}\mu(A')\min_{x\in A'}\bar{\varphi}_A(x)^2\max_{x:\|2^k\bar{c}(x-b_A)\|\geq 1}\frac{\|2^k\bar{c}(x-b_A)\|^{-2\tilde{d}}}{(\|2^k\bar{c}(x-b_A)\|+\bar{c}\|\mathbf{1}\|)^{-2\tilde{d}}}$$

$$\leq C'\sum_{A'\in\mathcal{A}}\mu(A')\min_{x\in A'}\bar{\varphi}_A(x)^2(1+\bar{c}\sqrt{d})^{2\tilde{d}}$$

$$\leq C'(1+\bar{c}\sqrt{d})^{2\tilde{d}}\int_\Omega \bar{\varphi}_A(x)^2\mathrm{d}x.$$

The quantity $\int_\Omega \bar{\varphi}_A(x)^2\mathrm{d}x$ on the right-hand side can be evaluated as

$$\int_\Omega \bar{\varphi}_A(x)^2\mathrm{d}x \leq \int_{x:\|x-b_A\|\leq 2\sqrt{d}}\bar{\varphi}_A(x)^2\mathrm{d}x$$

$$\leq \int_{x:\|x-b_A\|\leq\bar{c}^{-1}2^{-k}}\bar{\varphi}_A(x)^2\mathrm{d}x + \int_{x:\bar{c}^{-1}2^{-k}<\|x-b_A\|\leq 2\sqrt{d}}\bar{\varphi}_A(x)^2\mathrm{d}x$$

$$\lesssim 2^{-kd} + C\bar{c}^{-2\tilde{d}}2^{-2k\tilde{d}}\int_{\bar{c}^{-1}2^{-k}\leq r\leq 2\sqrt{d}}r^{-2\tilde{d}}r^{d-1}\mathrm{d}r$$

$$\lesssim 2^{-kd} + 2^{-2k\tilde{d}}\max\{2^{k(2\tilde{d}-d)},1\}$$

$$\lesssim \max\{2^{-kd},2^{-2k\tilde{d}}\}.$$

Therefore, we have that, for a constant $C''$, on the event $\mathcal{E}$, we have that

$$\frac{1}{n}\sum_{i=1}^n \varphi_A(x_i)^2 \leq C''(2^{-kd}\vee 2^{-2k\tilde{d}}).$$

Let $\mathcal{F}^\circ = \{\Delta\varphi_A \mid A \in \mathcal{A}\}$, then $\mathcal{F}^\circ$ satisfies Condition B. When $\tilde{d} \geq d/2$, by choosing $k$ so that $\bar{c}2^k \simeq n^{\frac{1}{2(\beta^*+d-\tilde{d}/p)}}$ and $K = K' = dk$, then Eq. (17) yields

$$R_*^{\mathrm{lin}}(\mathcal{F}^\circ) \gtrsim n^{-\frac{2(\beta^*-\tilde{d}/p+d/2)}{2(\beta^*-\tilde{d}/p+d/2)+d}}.$$

This concludes the proof.

**(ii-b)** *Setting of $\tilde{d} < d/2$:*

Let $\mathcal{A}$ be the partition of $\Omega$ as defined in the proof for $\tilde{d} \geq d/2$, i.e., $|\mathcal{A}| = 2^{dk}$ and each $A \in \mathcal{A}$ can be written as $A = [2^{-k}j_1, 2^{-k}(j_1+1)] \times \cdots \times [2^{-k}j_d, 2^{-k}(j_d+1)]$ for $0 \leq j_i \leq 2^k - 1$ $(i = 1, \ldots, d)$. Pick up $A \in \mathcal{A}$ and let $j \in \hat{J}(k)$ be the index such that $A = A_j$. For a while, we fix $A$ and let $\bar{b} = b_{A_j} = (2^{-k}(j_1+1/2), \ldots, 2^{-k}(j_d+1/2))^\top$ accordingly. For $\theta = (w, b) \in \mathbb{R}^{d-\tilde{d}+1}\times\mathbb{R}$, let $A_\theta : \mathbb{R}^d \to \mathbb{R}^{\tilde{d}}$ be

$$A_\theta(x) := 2^k[x_1 - \bar{b}_1, \ldots, x_{\tilde{d}-1} - \bar{b}_{\tilde{d}-1}, w^\top(x_{\tilde{d}:d} - \bar{b}_{\tilde{d}:d}) + b],$$

and consider

$$\phi_\theta(x) := \phi_{\tilde{d}}(A_\theta(x)).$$

We take its convex hull with respect to $\theta$. We note that

$$\phi_\theta(x) = \left(\prod_{j=1}^{\tilde{d}-1}\mathcal{N}_m(2^k(x_j - \bar{b}_j) - (m+1)/2)\right)\mathcal{N}_m\left(2^k[w^\top(x_{\tilde{d}:d} - \bar{b}_{\tilde{d}:d}) + b] - (m+1)/2\right).$$

To analyze its convex hull, it suffices to consider the convex hull of the last term $\mathcal{N}_m\left(2^k[w^\top(x_{\tilde{d}:d} - \bar{b}_{\tilde{d}:d}) + b] - (m+1)/2\right)$. Hence, we set $\psi(\cdot) := \mathcal{N}_m(\cdot - (m+1)/2)$ and consider a set of functions

$$\tilde{\mathcal{F}}_{C,\tau}^{(\psi)} := \{x \in \mathbb{R}^{d-\tilde{d}+1} \mapsto a\psi(\tau(w^\top x + b))) \mid |a| \le 2C,\ \|w\| \le 1,\ |b| \le 2\ (a, b \in \mathbb{R},\ w \in \mathbb{R}^{d-\tilde{d}+1})\}$$

for $C > 0$, $\tau > 0$. We also define the Fourier transform of $\psi$ as $\hat{\psi}(\omega) := (2\pi)^{-1}\int e^{-\mathrm{i}\omega x}\psi(x)\mathrm{d}x$ ($\omega \in \mathbb{R}$). Then, by Lemma 5, we have that, for $h = 2^{-k}$ and $\tau = h^{-1-\kappa}$,

$$\inf_{\check{g} \in \overline{\mathrm{conv}}(\tilde{\mathcal{F}}_{C,\tau}^{(\psi)})} \sup_{x \in [0,1]^d} \left| \check{g}(x) - \exp\left(-\frac{\|x - c\|^2}{2h^2}\right) \right|$$
$$\le \frac{4}{|2\pi\hat{\psi}(1)|}\left[C_{d-\tilde{d}+1}R^{2(d-\tilde{d}-1)}\exp(-R^2/2) + \exp(-R)\right],$$

where $C = \frac{\tau}{\pi|\hat{\psi}(1)|} = \Theta(h^{-1-\kappa})$ and $R = h^{-\kappa}(2\sqrt{d}+1)$. This indicates that, for a fixed $A \in \mathcal{A}$, the convex hull of the set $\{a\phi_\theta \mid \theta = (w, b) \in \mathbb{R}^{d-\tilde{d}+1} \times \mathbb{R},\ \|w\| \le 1,\ |b| \le 2,\ |a| \le \Delta\}$ where $\Delta = 2^{-k(\beta^* - \tilde{d}/p)}$ contains $\varphi_A$ which satisfies

$$\left\| \varphi_A - \Delta(2C)^{-1}\left(\prod_{j=1}^{\tilde{d}-1}\mathcal{N}_m(2^k(x_j - \bar{b}_j) - (m+1)/2)\right)\exp\left(-\frac{\|x_{\tilde{d}:d} - \bar{b}_{\tilde{d}:d}\|^2}{2h^2}\right)\right\|_\infty$$
$$= O\left(\Delta 2^{-k(1+\kappa)}(h^{-\kappa(2(d-\tilde{d}-1))}\exp(-h^{-2\kappa}/2) + \exp(-h^{-\kappa}))\right).$$

We can see that on the event $\mathcal{E}$, it holds that

$$\frac{1}{n}\sum_{i=1}^n \varphi_A^2(x_i) \lesssim \mu(A)(\Delta 2^{-k(1+\kappa)})^2 \lesssim 2^{-kd}2^{-2k(\beta^* - \tilde{d}/p+1)}2^{-2k\kappa} = 2^{-2k(\beta^* - \tilde{d}/p+1+d/2)-2k\kappa},$$

for all $A \in \mathcal{A}$. Let $\mathcal{F}^\circ = \{\varphi_A \mid A \in \mathcal{A}\}$, then $\mathcal{F}^\circ$ satisfies Condition B. Note that, by the definition of $\tilde{\mathcal{F}}_{C,\tau}^{(\psi)}$ is holds that $\varphi_A \in \overline{\mathrm{conv}}(\mathcal{H}_{\mathrm{aff}})$ for all $A \in \mathcal{A}$. Thus

$$R_*^{\mathrm{lin}}(\mathcal{H}_{\mathrm{aff}}) = R_*^{\mathrm{lin}}(\overline{\mathrm{conv}}(\mathcal{H}_{\mathrm{aff}})) \ge R_*^{\mathrm{lin}}(\mathcal{F}^\circ).$$

Therefore, by choosing $k$ such that $2^k \simeq n^{\frac{1}{2(\beta^* - \tilde{d}/p+1+d/2)+d+2\kappa}}$, and setting $K = K' = dk$, then Eq. (17) gives

$$R_*^{\mathrm{lin}}(\mathcal{F}^\circ) \gtrsim n^{-\frac{2(\beta^* - \tilde{d}/p+1+\kappa+d/2)}{2(\beta^* - \tilde{d}/p+1+\kappa+d/2)+d}}.$$

$\square$

**Lemma 5** (Suzuki & Akiyama (2021)). *Let $h > 0$ and $R := h\tau/(2\sqrt{d}+1)$. Then, for $C = \frac{\tau}{\pi|\hat{\psi}(1)|}$, the Gaussian RBF kernel can be approximated by*

$$\inf_{\check{g} \in \overline{\mathrm{conv}}(\tilde{\mathcal{F}}_{C,\tau}^{(\psi)})} \sup_{x \in [0,1]^d} \left| \check{g}(x) - \exp\left(-\frac{\|x - c\|^2}{2h^2}\right) \right|$$
$$\le \frac{4}{|2\pi\hat{\psi}(1)|}\left[C_d R^{2(d-2)}\exp(-R^2/2) + \exp(-R)\right]$$

*for any $c \in [0,1]^d$, where $C_d$ is a constant depending only on $d$. In particular, the right hand side is $O(\exp(-n^\kappa))$ if $R = n^\kappa$.*

## G Auxiliary lemmas

The following proposition which were shown in Schmidt-Hieber (2018); Hayakawa & Suzuki (2019); Suzuki (2018) is convenient to show the estimation error rate.

**Proposition 4** (Schmidt-Hieber (2018); Hayakawa & Suzuki (2019)). *Let $\mathcal{F}$ be a set of functions. Let $\widehat{f}$ be the least-squares estimator in $\mathcal{F}$:*

$$\widehat{f} = \underset{f \in \mathcal{F}}{\arg\min} \sum_{i=1}^{n} (y_i - f(x_i))^2.$$

*Assume that $\|f^{\circ}\|_{\infty} \leq F$ and all $f \in \mathcal{F}$ satisfies $\|f\|_{\infty} \leq F$ for some $F \geq 1$. If $\delta > 0$ satisfies $\mathcal{N}(\delta, \mathcal{F}, \|\cdot\|_{\infty}) \geq 3$, then it holds that*

$$\mathrm{E}_{D_n}[\|\widehat{f} - f^{\circ}\|_{L^2(P_X)}^2] \leq C \left[ \inf_{f \in \mathcal{F}} \|f - f^{\circ}\|_{L^2(P_X)}^2 + (F^2 + \sigma^2) \frac{\log \mathcal{N}(\delta, \mathcal{F}, \|\cdot\|_{\infty})}{n} + \delta(F + \sigma) \right],$$

*where $C$ is a universal constant.*

The following lemma provides the covering number of the deep neural network model.

**Lemma 6** (Covering number evaluation). *The covering number of $\Phi(L, W, S, B)$ can be bounded by*

$$\log \mathcal{N}(\delta, \Phi(L, W, S, B), \|\cdot\|_{\infty}) \leq S \log(\delta^{-1} L (B \vee 1)^{L-1} (W+1)^{2L})$$
$$\leq 2SL \log((B \vee 1)(W+1)) + S \log(\delta^{-1} L).$$

*Proof of Lemma 6.* Given a network $f \in \Phi(L, W, S, B)$ expressed as
$$f(x) = (\mathcal{W}^{(L)} \eta(\cdot) + b^{(L)}) \circ \cdots \circ (\mathcal{W}^{(1)} x + b^{(1)}),$$
let
$$\mathcal{A}_k(f)(x) = \eta \circ (\mathcal{W}^{(k-1)} \eta(\cdot) + b^{(k-1)}) \circ \cdots \circ (\mathcal{W}^{(1)} x + b^{(1)}),$$
and
$$\mathcal{B}_k(f)(x) = (\mathcal{W}^{(L)} \eta(\cdot) + b^{(L)}) \circ \cdots \circ (\mathcal{W}^{(k)} \eta(x) + b^{(k)}),$$
for $k = 2, \ldots, L$. Corresponding to the last and first layers, we define $\mathcal{B}_{L+1}(f)(x) = x$ and $\mathcal{A}_1(f)(x) = x$ respectively. Then, it is easy to see that $f(x) = \mathcal{B}_{k+1}(f) \circ (\mathcal{W}^{(k)} \cdot + b^{(k)}) \circ \mathcal{A}_k(f)(x)$. Now, suppose that a pair of different two networks $f, g \in \Phi(L, W, S, B)$ given by

$$f(x) = (\mathcal{W}^{(L)} \eta(\cdot) + b^{(L)}) \circ \cdots \circ (\mathcal{W}^{(1)} x + b^{(1)}), \quad g(x) = (\mathcal{W}^{(L)'} \eta(\cdot) + b^{(L)'}) \circ \cdots \circ (\mathcal{W}^{(1)'} x + b^{(1)'}),$$

has parameters with distance $\delta$: $\|\mathcal{W}^{(\ell)} - \mathcal{W}^{(\ell)'}\|_{\infty} \leq \delta$ and $\|b^{(\ell)} - b^{(\ell)'}\|_{\infty} \leq \delta$. Now, not that $\|\mathcal{A}_k(f)\|_{\infty} \leq \max_j \|\mathcal{W}_{j,:}^{(k-1)}\|_1 \|\mathcal{A}_{k-1}(f)\|_{\infty} + \|b^{(k-1)}\|_{\infty} \leq WB \|\mathcal{A}_{k-1}(f)\|_{\infty} + B \leq (B \vee 1)(W+1) \|\mathcal{A}_{k-1}(f)\|_{\infty} \leq (B \vee 1)^{k-1}(W+1)^{k-1}$, and similarly, the Lipshitz continuity of $\mathcal{B}_k(f)$ with respect to $\|\cdot\|_{\infty}$-norm is bounded as $(BW)^{L-k+1}$. Then, it holds that

$$|f(x) - g(x)|$$
$$= \left| \sum_{k=1}^{L} \mathcal{B}_{k+1}(g) \circ (\mathcal{W}^{(k)} \cdot + b^{(k)}) \circ \mathcal{A}_k(f)(x) - \mathcal{B}_{k+1}(g) \circ (\mathcal{W}^{(k)'} \cdot + b^{(k)'}) \circ \mathcal{A}_k(f)(x) \right|$$
$$\leq \sum_{k=1}^{L} (BW)^{L-k} \|(\mathcal{W}^{(k)} \cdot + b^{(k)}) \circ \mathcal{A}_k(f)(x) - (\mathcal{W}^{(k)'} \cdot + b^{(k)'}) \circ \mathcal{A}_k(f)(x)\|_{\infty}$$
$$\leq \sum_{k=1}^{L} (BW)^{L-k} \delta [W(B \vee 1)^{k-1}(W+1)^{k-1} + 1]$$
$$\leq \sum_{k=1}^{L} (BW)^{L-k} \delta (B \vee 1)^{k-1} (W+1)^k \leq \delta L (B \vee 1)^{L-1} (W+1)^L.$$

Thus, for a fixed sparsity pattern (the locations of non-zero parameters), the covering number is bounded by $\left(\delta / [L(B \vee 1)^{L-1}(W+1)^L]\right)^{-S}$. There are the number of configurations of the sparsity pattern is bounded by $\binom{(W+1)^L}{S} \leq (W+1)^{LS}$. Thus, the covering number of the whole space $\Phi$ is bounded as

$$(W+1)^{LS} \left\{ \delta / [L(B \vee 1)^{L-1}(W+1)^L] \right\}^{-S} = [\delta^{-1} L (B \vee 1)^{L-1} (W+1)^{2L}]^S,$$

which yields the assertion.

$\square$