# OpenReview forum: "Deep learning is adaptive to intrinsic dimensionality of model smoothness in anisotropic Besov space"
_NeurIPS.cc/2021/Conference — NeurIPS 2021 Spotlight_

### Official Review · Reviewer_p9g3 · 2021-07-16

**Rating:** 7
**Confidence:** 3

**Summary:**

In this paper, the authors investigate the approximation and estimation error of deep neural networks on anisotropic Besov spaces. It has been shown that Holder and Sobolev function spaces are plagued by an unavoidable curse of dimensionality, and a line of research has focused on showing how deep neural network can adaptively beat this curse on restricted classes of functions, such as compositorial functions or functions that depend on a low dimension subspace. This work focuses on Besov spaces, but with anisotropic smoothness requirements (which do not require for the data to lie exactly on a low dimensional subspace) and consider deep composition model similarly to Schmidt-Hieber (2018), but with functions in the anisotropic Besov spaces (they don’t need to be low-dimensional). They show that the rate of approximation/estimation depends on the harmonic mean of the directional smoothness parameters, instead of the worst case. In particular, this harmonic mean is independent of the dimension if the function is non smooth in only a few directions. Their results require bounds on the width, depth, sparsity, and infinity norm of the weights. They further show that the estimation error rates are minmax optimal. Finally they consider linear estimators and show a lower bound on their minimax rate, that is larger than the minimax rate of neural networks, which illustrates the poor adaptivity of linear estimators to the intrinsic dimensionality of the data.

**Limitations And Societal Impact:**

Yes.

**Main Review:**

This paper present new strong results which combines expressivity and generalization bounds on deep neural networks. The adaptivity to the average smoothness instead of the worst case smoothness is particularly interesting. For these reasons, I recommend the paper to be accepted.

It would be great to comment more on the dependency of the different quantities on $n,d$ and other parameters. The current theorems are difficult to parse. For example, $N$ is proportional to the number of non zero parameters? For $\tilde{\beta}$ to not be of order $1/d$, $m$ needs to scale with $d$, is $L_1(d)$ linear in $d$ then? I think adding a few remarks would go a long way in improving the readability of these results.

While I think this represents an interesting theoretical work for the approximation/function space estimation community, I wonder how much this approach can help understand practical success of deep learning (in particular, I disagree with the last sentence of the conclusion ``''these analyses strongly support the practical success of deep learning from a theoretical perspective’’). In particular, numerous work have demonstrated that one cannot disentangle computational and statistical aspects in deep learning. Furthermore, the theorems have a sparsity requirement, which is usually not the case for practically trained neural networks.

Typos: reference to Theorem 6 in Theorem 2? Reference to equation 8 bottom of page 6?


-------- update after rebuttal -------

Thank you to the authors for the response. I am satisfied with the paper and the updates mentioned by the authors. I keep the same rating.

**Time Spent Reviewing:**

3

---

> ### Author Response · Authors · 2021-08-05
> **Reply to reviewer p9g3**
>
> Thank you very much for your encouraging comments.
>
> **Q:** It would be great to comment more on the dependency of the different quantities on n,d and other parameters. The current theorems are difficult to parse. For example, N is proportional to the number of non zero parameters? For
> \tilde{\beta} to not be of order 1/d, m needs to scale with d, is L_1(d) linear in d then? I think adding a few remarks would go a long way in improving the readability of these results.
> **A:** Yes, $N$ is approximately the number of non-zero parameters. More precisely, the number of non-zero parameters $S$ satisfies $S = N\log(N)$ because $L = O(\log(N))$. As for the dependency on the dimensionality $d$, the depth $L_1(d)$ is proportional to $d$. This is because we used a tensor product B-spline to evaluate the expressive ability of deep neural networks and then there appears $d$-times multiplications in the analysis resulting in $O(d)$-depth. The influence of $d$ to the rate of convergence is more involved because a hidden constant in the order symbol has complicated dependency on $d$. Therefore, our bound assumes $d$ is a constant and does not allow to increase as the parameter $N$ or the sample size $n$ goes to infinity. We would like to defer it to another work to deal with a situation where $d$ goes to infinity. Anyway, we would like to add these kind of discussions to the main text as remarks.
>
>
> **Q:** While I think this represents an interesting theoretical work for the approximation/function space estimation community, I wonder how much this approach can help understand practical success of deep learning (in particular, I disagree with the last sentence of the conclusion ``''these analyses strongly support the practical success of deep learning from a theoretical perspective’’). In particular, numerous work have demonstrated that one cannot disentangle computational and statistical aspects in deep learning. Furthermore, the theorems have a sparsity requirement, which is usually not the case for practically trained neural networks.
> **A:** Indeed, our theoretical contribution would not be directly connected to the practical application. We believe that our analysis partially explain the practical usefulness of deep learning. We agree that the computational aspects are big issues for this type of theoretical analysis, but we also think that it would be a proper procedure to develop theory in different aspect step-by-step and we expect that a future work on computational complexity will fill the gap. As for the sparsity, we also agree that it would be a bit non-practical. However, [R1] showed that a dense network can achieve the optimal rate to approximate functions in the Holder space. We think that this technique can be applied to our setting, anisotropic Besov space. We would like to defer this issue to the future work.
>
> [R1] Jianfeng Lu, Zuowei Shen, Haizhao Yang, Shijun Zhang: Deep Network Approximation for Smooth Functions. https://arxiv.org/abs/2001.03040.
>
>
> **Q:** Typos: reference to Theorem 6 in Theorem 2? Reference to equation 8 bottom of page 6?
> **A:** We are sorry that Theorem 6 and equation 8 are in the appendix deferred to the supplementary material. We will add a sentence to clarify this point, like "Theorem 6 in the supplementary material" and "Eq.(8) in the supplementary material."

---

### Official Review · Reviewer_PyLN · 2021-07-16

**Rating:** 7
**Confidence:** 4

**Summary:**

This paper provides a constructive approximation results of deep neural networks for approximating functions in anisotropic Besov spaces. The approximation result is further used to show the statistical convergence of using neural networks in a nonparametric regression problem. The highlights of the work fall in the circumvent of the curse of data dimensionality, when the anisotropic Besov space is not homogeneous in every canonical direction.

Some recent works study the adaptivity of neural networks to the intrinsic structures in data (Nakada & Imaizumi, 2020; Schmidt-Hieber, 2019; Chen et al., 2019), which can partially explain the practical success of neural networks in high-dimensional data applications. This work takes a different perspective: rather than considering the low-dimensional structures in the data manifold, the authors focus on "low-dimensional" structures in the function space. Intuitively, the "low-dimensional" structures in the function space refers to functions having very mild fluctuations in certain directions. This is a relatively less explored direction, yet of importance.

The paper is well organized and relatively easy to follow.

**Limitations And Societal Impact:**

The authors briefly discussed the limitations in Section 7. No direct negative societal impact.

**Main Review:**

I am tending positive on the paper. There are a few questions.

1) Is it true that the composition of two anisotropic Besov functions is still an anisotropic Besov function? If so, I don't see the merit of considering two models of the true functions in Section 2.2, as deep composition can also be viewed as a special case of affine composition model, and vice versa.

2) Per definition of $\tilde{\beta}$ in Equation (1), when $\beta_j$ is approximately equal, there is still the curse of data dimensionality. However, when only a small fraction of $\beta_j$'s are small (severe fluctuation of the target function in these directions), the result is free of the curse. In such a case, is it possible that the target function may be approximated first by a function only supported on those significant coordinates, and then approximate the function on those significant coordinates? It seems that this argument translates finally to some low-dimensional structures in data. Can the authors kindly comment on this?

3) A practical motivation of studying anisotropic Besov spaces seems missing.

**Time Spent Reviewing:**

12 hours

---

> ### Author Response · Authors · 2021-08-05
> **Reply to reviewer PyLN**
>
> Thank you very much for your encouraging comments.
>
> **Q:** Is it true that the composition of two anisotropic Besov functions is still an anisotropic Besov function?
> **A:** No, a composition of two anisotropic Besov functions are generally not in an anisotropic Besov space. For example, composition of a smooth nonlinear transformation and anisotropic Besov function is not in an anisotropic Besov space. It could be included in an anisotropic Besov with much smaller smoothness \beta, but such interpretation gives much slower convergence rate which could be far from a precise evaluation. Therefore, estimating compositions is a harder problem than estimating a single anisotropic Besov function.
>
> **Q:** Per definition of in Equation (1), when  is approximately equal, there is still the curse of data dimensionality. However, when only a small fraction of 's are small (severe fluctuation of the target function in these directions), the result is free of the curse. In such a case, is it possible that the target function may be approximated first by a function only supported on those significant coordinates, and then approximate the function on those significant coordinates?
> **A:** Yes, that is possible. This is the part of the motivation why we considered the affine composition model. If we use the affine composition model, then we can do as you suggested. Here, we would like to point out that even in such a situation, the dimensionality of the support of data distribution is still high dimensional. Therefore, we need a low dimensional structure on the function space.
>
>
> **Q:** A practical motivation of studying anisotropic Besov spaces seems missing.
> **A:** In practice, we think anisotropic smoothness is much more natural than isotropic smoothness because the latter one imposes "precisely" the same sensitivity toward every direction. However, features typically have different sensitivity to the output and thus anisotropic smoothness seems more natural. One example is given in line 54-55, that is, the true function is likely to invariant against perturbations of an input in some specific directions (Figure 1). Another example is that, considering a function which takes image as an input, an image can be decomposed into different frequency components and usually a function of image has less sensitivity on the high frequency components and more dependent on the low frequency components. As this example indicates, we consider that the anisotropic smoothness is much more typical in practice than the isotropic smoothness. We would like to add this point in the final version.

---

> > ### Comment · Reviewer_PyLN · 2021-08-21
> > **Thank you for clarification. No change to the score.**
> >
> > The author's response cleared my concerns, and thank you for pointing out the composition of two anisotropic Besov functions can undermine the smoothness parameter $\beta$.
> >
> > I keep my score.

---

### Official Review · Reviewer_DZ84 · 2021-07-17

**Rating:** 7
**Confidence:** 4

**Summary:**

The paper studies adaptivity of deep neural network models to anisotropic smoothness properties of target functions. Namely, when the target function belongs to an anisotropic Besov space (i.e. it has different levels of smoothness in different input directions), certain classes of deep networks may provide a rate that only depends on the average smoothness (i.e. the harmonic mean of certain smoothness parameters in each direction), instead of the worst-case smoothness. This is also studied for target functions that consist of compositions of such anisotropic Besov functions and compositions with an affine function. The authors also provide minimax lower bounds showing that linear estimators (such as kernel methods) may not adapt to such structures, and may provide exponentially worse sample complexity for some specific models.

**Limitations And Societal Impact:**

yes

**Main Review:**

The paper studies adaptivity properties of deep networks, a topic which has recently gained a lot of attention, as it can yield an understanding of what kind of functions deep learning can learn efficiently, and more efficiently than other standard procedures such as kernels. It provides a comprehensive study of adaptivity of neural networks to anisotropic smoothness, which is a useful prior, and has not been previously studied in this context. In this sense, I find the paper significant and novel. It is also very well written. I thus recommend acceptance.

A few minor comments:
* the authors briefly mention a motivation for anisotropic smoothness in the context of invariances on image problems. This seems quite interesting but the current explanation is very brief - I encourage the authors to expand further on this, and perhaps on other motivating examples.
* why is tilde{beta} in (1) defined like this instead of just as the harmonic mean? I find that this makes the rates with +1 instead of +d a bit puzzling to parse, and would find it more natural to just have the harmonic mean, so that you immediately recognize the usual rate when all the betas are equal (but I leave this choice to the authors).
* some typos: L199: d beta -> beta / d? L301: why the red text?


-------- update after rebuttal -------

Thank you for the response. I think these additional comments on motivating examples would be useful in the paper, so please do include them in the final version.

**Time Spent Reviewing:**

3

---

> ### Author Response · Authors · 2021-08-05
> **Reply to reviewer DZ84**
>
> Thank you very much for your encouraging comments. Please find our responses to your comments in the following.
>
> **Q:** I encourage the authors to expand further on this, and perhaps on other motivating examples.
> **A:** Thank you very much for your constructive comment. Due to the space limitation, we should have saved detailed explanations about practical aspects.  Another example could be that, considering a function which takes an image as an input, an image can be decomposed into different frequency components and usually a function of images has less sensitivity on the high frequency components and more dependent on the low frequency components which yields anisotropic smoothness. As this example indicates, we consider that the anisotropic smoothness is much more typical in practice than the isotropic smoothness which requires "completely same" sensitivity toward all directions. We would like to enhance this point in the final version for which we would be allowed to use more space.
>
>
> **Q:** why is tilde{beta} in (1) defined like this instead of just as the harmonic mean?
> **A:** This is because we wanted to tread \tilde{\beta} as a single quantity that represents the complexity of the model. We consider that it is more essential to use \beta/d as a complexity measure of the usual Besov space instead of separating \beta and d.
>
> **Q:** some typos: L199: d beta -> beta / d?
> **A:** Thank you very much for pointing out it. You are absolutely right. We will fix this in the final version.
>
>
> **Q:** L301: why the red text?
> **A:** We are sorry that this was a bit confusing. We guess you used the document in the supplementary material. The red color indicates a difference from the main text in which the \kappa term was not explicitly included.

---

### Decision · Program_Chairs · 2021-09-27

**Decision:**

Accept (Spotlight)

**Comment:**

This is a theory paper that analyzes deep neural networks through the lens of anisotropic Besov space. The paper provides a comprehensive study of adaptivity of neural networks to anisotropic (=direction-dependent) smoothness. The paper is clearly written and the authors have carefully positioned their work relative to prior works.